# PHYSIOME-ODE: A BENCHMARK FOR IRREGULARLY SAMPLED MULTIVARIATE TIME SERIES FORECASTING BASED ON BIOLOGICAL ODES

**Christian Klötergens**
ISMLL & VWFS DARC
University of Hildesheim
Hildesheim, Germany
`kloetergens@ismll.de`

**Vijaya Krishna Yalavarthi**
ISMLL
University of Hildesheim
Hildesheim, Germany

**Randolf Scholz**
ISMLL
University of Hildesheim
Hildesheim, Germany

**Maximilian Stubbemann**
ISMLL & VWFS DARC
University of Hildesheim
Hildesheim, Germany

**Stefan Born**
Institute of Mathematics
TU Berlin
Berlin, Germany

**Lars Schmidt-Thieme**
ISMLL & VWFS DARC
University of Hildesheim
Hildesheim, Germany

## ABSTRACT

State-of-the-art methods for forecasting irregularly sampled time series with missing values predominantly rely on just four datasets and a few small toy examples for evaluation. While ordinary differential equations (ODE) are the prevalent models in science and engineering, a baseline model that forecasts a constant value outperforms ODE-based models from the last five years on three of these existing datasets. This unintuitive finding hampers further research on ODE-based models, a more plausible model family. In this paper, we develop a methodology to generate irregularly sampled multivariate time series (IMTS) datasets from ordinary differential equations and to select challenging instances via rejection sampling. Using this methodology, we create `Physiome-ODE`, a large and sophisticated benchmark of IMTS datasets consisting of 50 individual datasets, derived from ODE models developed by research in Biology. `Physiome-ODE` is the first benchmark for IMTS forecasting that we are aware of and an order of magnitude larger than the current evaluation setting. Using `Physiome-ODE`, we show qualitatively completely different results than those derived from the current four datasets: on `Physiome-ODE` deep learning methods based on ODEs can play to their strength and our benchmark can differentiate in a meaningful way between different IMTS forecasting models. This way, we expect to give a new impulse to research on irregular time series modeling.

## 1 INTRODUCTION

Over the past decade, substantial research has been devoted to fully observed and regularly sampled time series, often referred to as regular (multivariate) time series. However, in certain scenarios, observations occur at irregular intervals and variables are measured independently resulting in a sparse multivariate time series with many missing values. The resulting time series are known as irregularly sampled multivariate time series with missing values, or simply **IMTS**.

With regular time series forecasting being a well-established topic, there exist a variety of sophisticated methods and standardized benchmarks (Godahewa et al., 2021; Gilpin, 2021; Bauer et al., 2021). On the other hand, there is currently no IMTS forecasting benchmark published. Creating such a benchmark demands a high number of publicly available time series datasets with genuinely irregular observation processes. However, only a few of those exist. Although they are closely related to each other, most evaluation sections of IMTS forecasting papers (Biloš et al., 2021; Schirmer et al., 2022; Yalavarthi et al., 2024; Zhang et al., 2024) are based on at least a subset of these datasets. Hence, it stands to reason that they induce a dataset bias. Additionally, we show that a simple baseline,

which always predicts a constant value independent of time, is competitive with or even outperforms complex neural ODE models on these datasets. Hence, it appears questionable whether the currently used datasets are indeed well-suited for forecasting.

To address this limitation, we introduce `Physiome-ODE`, a wide benchmark of IMTS datasets consisting of 50 individual datasets, derived from ordinary differential equations from biological research, that are stored in the Physiome Model Repository (PMR). Biological processes are well-suited for generating IMTS datasets, as they are inherently multivariate and irregularly measured in real-world experiments. Additionally, the PMR provides Python implementations of many of these models, allowing us to create `Physiome-ODE` in an automated manner. While Biology researchers create their models based on very few and non-published observations, they enable us to create an arbitrary number of time series which relate to possible measurements of a real-world phenomenon. `Physiome-ODE` is the first benchmark for IMTS forecasting that we are aware of and an order of magnitude larger than the current evaluation setting of just four datasets.

Furthermore, to evaluate the complexity of the different forecasting datasets, we introduce a simple metric called *Joint Gradient Deviation (*JGD*)*, which measures the gradient variance of ODE solutions. We will show that our benchmark consists of datasets of different complexity and covers a wide range of different JGD values.

Finally, we evaluate current IMTS forecasting methods on `Physiome-ODE` and show that it includes many datasets on which neural ODE-based models significantly outperform the time-constant baseline model. Furthermore, a member of the neural ODE model family actually emerges as the overall most accurate model on `Physiome-ODE`. However, the datasets in `Physiome-ODE` are diverse enough that no single model is the most accurate for every dataset.

`Physiome-ODE` is a significant step forward to standardized IMTS forecasting and to monitor research progress. Our contributions include the following:

1. We introduce a simple baseline model that is restricted to making constant forecasts. On traditional IMTS forecasting evaluation, this simple baseline shows competitive or even better forecasting accuracy when compared to ODE-based models.

2. We propose *Joint Gradient Deviation (*JGD*)*, a simple score designed to approximate the difficulty of time series datasets for forecasting.

3. We create `Physiome-ODE`, a large and diverse benchmark of IMTS forecasting datasets. Maximizing JGD, we select and configure ODE models present in the Physiome (Yu et al., 2011) repository. `Physiome-ODE` is the first benchmark for IMTS forecasting that we are aware of.

4. We evaluate state-of-the-art IMTS forecasting models on `Physiome-ODE`. Our experiments show that the datasets included in `Physiome-ODE` are diverse enough to highlight different strengths of competing models. This results in different performance rankings of competing models for each dataset and the absence of a single best model. In that, `Physiome-ODE` differs from the currently used IMTS forecasting datasets, which is expected to provide a fresh impulse to research on IMTS forecasting models.

5. We share our code on GitHub (https://github.com/kloetergensc/Physiome-ODE)

## 2 IRREGULARLY SAMPLED TIME SERIES FORECASTING

An *irregularly sampled multivariate time series (IMTS)* is a sequence $X := ((t_i, v_i))_{i=1:I} \in (\mathbb{R} \times (\mathbb{R} \cup \{\texttt{NaN}\})^C)^* =: \mathcal{X}$ of pairs where $t_i$ is the observation timepoint and $v_i$ is an observation event with maximum of $C$-many variables (or channels) being observed. We use $^*$ to indicate the space of sequences. We call $v_{i,c} \in \mathbb{R}$ an observed value and $v_{i,c} = \texttt{NaN}$ a missing value. $Q := (t_k^{\text{QRY}}, c_k^{\text{QRY}})_{k=1:K} \in (\mathbb{R} \times \{1, \dots, C\})^* =: \mathcal{Q}$ is the query sequence where $t_k^{\text{QRY}}$ is a queried (future) timepoint and $c_k^{\text{QRY}}$ is a queried channel. $Y := (y_1, \dots, y_K) \in \mathbb{R}^* =: \mathcal{Y}$ is the forecasting answer where $y_k$ is the ground truth forecast value for $Q_k$.

Then the *IMTS forecasting problem* is defined as follows: given a sequence of triples $((X_n, Q_n, Y_n))_{n=1:N}$ drawn from a random distribution $\rho$ where $X_n \in \mathcal{X}$ is an irregularly sampled

Table 1: Test MSE for forecasting next 50% after 50% observation time. OOM refers to out of memory. We highlight the best model in **bold** and underline the second best. [†] indicates that we show the results from Yalavarthi et al. (2024)

| Model | USHCN | PhysioNet-2012 | MIMIC-III | MIMIC-IV |
|---|---|---|---|---|
| GRU-ODE | $1.017\pm0.325$ | $0.653\pm0.023$[†] | $0.653\pm0.023$[†] | $0.439\pm0.003$[†] |
| LinODEnet | $0.662\pm0.126$ | $0.411\pm0.001$[†] | $0.531\pm0.022$[†] | $0.336\pm0.002$[†] |
| CRU | $0.730\pm0.264$ | $0.467\pm0.002$[†] | $0.619\pm0.028$[†] | OOM[†] |
| Neural Flow | $1.014\pm0.336$ | $0.506\pm0.002$[†] | $0.651\pm0.017$[†] | $0.465\pm0.003$[†] |
| GraFITi | **$0.636\pm0.161$** | **$0.401\pm0.001$**[†] | **$0.491\pm0.014$**[†] | **$0.285\pm0.002$**[†] |
| GraFITi-C | $0.875\pm0.204$ | $0.407\pm0.001$ | $0.543\pm0.024$ | $0.324\pm0.002$ |

time series, $Q_n \in \mathcal{Q}$ is a query sequence and $Y_n \in \mathcal{Y}$ is a ground truth answer, as well as a loss function $\ell : \mathcal{Y} \times \mathcal{Y} \to \mathbb{R}$, e.g., the mean squared error between two equilength sequences, the task is to find a model $\hat{Y} : \mathcal{X} \times \mathcal{Q} \to \mathcal{Y}$ such that the expected loss between the ground-truth forecasting answer $Y$ and the model prediction $\hat{Y}(X, Q)$ is minimal:

$$\mathcal{L}(\hat{Y}; \rho) := \mathbb{E}_{(X,Q,Y)\sim\rho} \, \ell(Y, \hat{Y}(X, Q)). \tag{1}$$

## 2.1 Current Forecasting Models

In our experiments we compare five recent models. Four of these five network architecture are variants of the neural ODE (Chen et al., 2018), which uses an ODE, that is defined by a neural network, to infer the hidden state in-between observations. At observation times the neural ODE-based estimation of the hidden state is updated on the actual observation. For more details, we refer to the papers. We use the following models:

**GRU-ODE-Bayes** (De Brouwer et al., 2019) utilizes a continuous version of Gated Recurrent Units (GRU) as the ODE defining network $f$. Additionally, GRU-Bayes applies a GRU-based Bayesian update to the hidden state at observations. **LinODEnet** (Scholz et al., 2022) restricts $f$ to be linear functions but encodes data non-linearly into latent space. At observations LinODEnet updates its hidden state with a so-called nonlinear KalmanCell, a module inspired by Kalman Filtering (Kalman, 1960). **Continuous Recurrent Units** (Schirmer et al., 2022) replaces the neural ODE with a stochastic differential equation (SDE). CRU is able to infer the hidden state at any time in closed form using continuous-discrete Kalman filtering, instead of solving a neural ODE. **Neural Flows** (Biloš et al., 2021) infers the solution curve of an ODE directly with invertible neural networks. **GraFITi** (Yalavarthi et al., 2024) is substantially different from neural ODE. It encodes the observations from IMTS into a graph structure and infers forecasts by solving a graph completion problem with a graph attention network. On the established evaluation datasets GraFITi significantly outperforms the neural ODE-based models in terms of forecasting accuracy.

## 3 Time-Constant Approaches Outperform ODE-based Approaches in the Current Evaluation Scenario

**Current Evaluation Scenario.** Traditionally, IMTS forecasting has relied on datasets like PhysioNet2012 (Silva et al., 2012), MIMIC-III (Johnson et al., 2016), and MIMIC-IV (Johnson et al., 2021) obtained from Kaggle challenges designed for IMTS classification. Additionally, De Brouwer et al. (2019) proposed to create an IMTS forecasting task, based on USHCN (Menne et al., 2016), a fully observed regularly sampled time series dataset containing climate data. We present characteristics of these datasets in Table 3 (appendix).

The GraFITi model is the state-of-the-art on PhysioNet2012, MIMIC-III, MIMIC-IV, and USHCN, demonstrating a significant performance advantage over neural ODE-based methods (Yalavarthi et al., 2024). To further analyze this gap, we designed a *constant* version of GraFITi (GraFITi-C), which is restricted to make the exact same forecast for every query time point. This is achieved by training a GraFITi model that cannot access the actual query time point ($t_k^{\text{QRY}}$), but is instead always provided with a constant dummy query time point.

The results shown in Table 1 indicate, that neural ODE-based models are indeed outperformed by this allegedly easy-to-beat baseline on PhysioNet2012 and MIMIC-IV. On MIMIC-III the best performing ODE-based model achieves a lower average test MSE than GraFITi-C, however the difference is within the standard deviation. GraFITi on the other hand, consistently outperforms its constant variant. However, it also cannot separate in by a drastic margin. These findings could relate to important covariants, which are not contained in the dataset. For instance, it is impossible to predict a patient's blood glucose level without information about any food intake.

We designed `Physiome-ODE` to be an extension to the currently used evaluation protocol. For our ODE generated datasets, we can guarantee that all necessary covariants are given. Furthermore, we can take full control about the simulated observation process.

## 4 GENERATING CHALLENGING IMTS DATASETS FROM ODES

**Sampling ODEs to Generate IMTS Data.** In science and engineering, processes that evolve in continuous time are usually described by differential equations, in the simpler cases by ordinary differential equations (ODEs). All possible observations of such a process with $C$ **channels** over **duration** $T$ at any time $t$ can be captured by a time-varying function $x \colon [0, T] \to \mathbb{R}^C$ as $x(t)_c$. An ODE characterizes such a function implicitly by two conditions:

1. A relation between observation time $t$, observation value $x(t)$ and the instantaneous change in observation values $x'(t)$, i.e. the first derivative of $x$. This relation is described by a function $F : [0, T] \times \mathbb{R}^C \times \mathbb{R}^C \to \mathbb{R}$ called the **system**.

2. The **initial values** $x_0 \in \mathbb{R}^C$ of the function at time $t = 0$:

$$F(t, x(t), x'(t)) = 0 \quad \forall t \in [0, T], \qquad x(0) = x_0 \tag{2}$$

Note that ODEs can be defined including higher order derivatives, but w.l.o.g. one can assume they are of first order, since the higher order system can be transformed into a first order system by a change of variables (Teschl, 2012). In a few special cases the solution $x(t)$ can be computed analytically. For example, the one-dimensional system $F(t, x(t), x'(t)) := ax(t) - x'(t)$ characterizes the exponential growth function $x(t) = e^{at}x_0$. However, in most cases, it can only be computed by a numerical ODE solver.

Specific parameters that occur in a system $F$ like the growth factor $a$ in our example are called **constants** and one writes $F(t, x(t), x'(t); a)$ with $a \in \mathbb{R}^A$ to denote a system with $A$-many constants. Triples $(F, a, x_0)$ of systems, constants and initial values can be turned into a simulator, i.e. a generative model, by complementing them with two more components:

1. A **sampling process** $s$ for observation time points and channels that can generate finite sequences $((t_1, c_1), \dots, (t_I, c_I))$ of pairs of observation times $t_i \in [0, T]$ and observed channels $c_i \in \{1, \dots, C\}$.

2. A **noise model** $p_{\text{NOISE}}$ for the observation values, that is a conditional distribution $p_{\text{NOISE}}(x^{\text{OBS}} \mid x^{\text{TRUE}})$ of the observed value given the ground truth value.

Then one can generate irregular sampled instances $(t_i, c_i, x_i^{\text{OBS}})_{i=1:I}$ of the (only implicitly given) underlying function $x$ with missing values and affected by observation noise simply via:

$$\big((t_i, c_i)\big)_{i=1:I} \sim s \qquad x_i^{\text{TRUE}} := \text{ode-solve}(F, a, x_0, t_i)_{c_i} \qquad x_i^{\text{OBS}} \sim p_{\text{NOISE}}(x_i^{\text{OBS}} \mid x_i^{\text{TRUE}}) \tag{3}$$

**Identifying Challenging Time Series.** While any ODE implicitly describes a time-varying function $x \colon [0, T] \to \mathbb{R}$, not all such functions are challenging to forecast: For example, some just describe very simple convergence processes that approach a limit value over a long time. In order to discover challenging time series, we propose to measure the *mean gradient deviation* (MGD), that is the time-normalized $L^2$ norm of $x'(t)$ around its mean:

$$\text{MGD}(x) := \min_c \sqrt{\frac{1}{T} \int_0^T (x'(t) - c)^2 \, \mathrm{d}t} \tag{4}$$

Note that the minimum is obtained precisely for the mean gradient

$$c = \frac{1}{T} \int_0^T x'(t) \, dt = \frac{x(T) - x(0)}{T}.$$ (5)

For a linear test function $x(t) = at + b$, the MGD is zero, while for a sine wave $x(t) = \sin(\omega t)$, the MGD scales linearly with the frequency $\omega$. To identify not just individually challenging time series, but whole distributions (i.e. a generator $p$), one can compute the expectation of MGD. We found that the MGD grows super-exponentially with the Lipschitz constant, which is described in more detail in Appendix C

With MGD one does assess only the heterogeneity within each function, not heterogeneity between different functions, which is an important precondition for an interesting data set for machine learning tasks. To capture this second aspect, we propose the *mean point-wise gradient deviation* (MPGD):

$$\text{MPGD}(p) := \frac{1}{T} \int_0^T \text{std}_{x \sim p}[x'(t)] \, dt = \frac{1}{T} \int_0^T \sqrt{\mathbb{E}_{x \sim p}\left[\left(x'(t) - \mathbb{E}_{y \sim p}[y'(t)]\right)^2\right]} \, dt$$ (6)

We combine both measures multiplicatively as *joint gradient deviation* (JGD) for distributions of functions, i.e. function generators:

$$\text{JGD}(p) := \text{MPGD}(p) \cdot \mathbb{E}_{x \sim p}[\text{MGD}(x)]$$ (7)

Note, that we can approximate MGD with a finite number of samples:

**Lemma 1.** *For a function $x \in \mathcal{C}^1([0, T])$ and $\epsilon > 0$ such that $\frac{T}{\epsilon} \in \mathbb{N}$ we consider the divided differences*

$$x_t^{\text{diff}, \epsilon} := \frac{x(t) - x(t - \epsilon)}{\epsilon}, \quad t = \epsilon, 2\epsilon, \ldots, T.$$

*It then holds that the numerical estimator $\widehat{\text{std}}[(x_k)_{k=1:K}] := \sqrt{\frac{1}{K} \sum_{k=1}^{K} (x_k - \bar{x})^2}$ of the standard deviation of the divided differences converges to the MGD of the function:*

$$\widehat{\text{std}}[x^{\text{diff}, \epsilon}] \xrightarrow{\epsilon \to 0} \text{MGD}(x)$$ (8)

We can also approximate MPGD with a finite amount of samples under assumptions we explain in Appendix B.

**Lemma 2.** *The mean point-wise gradient deviation of a distribution $p$ of functions can be approximated on a sample $(x_n)_{n=1:N}$ of $N$-many sequence from $p$ on a fixed grid of time points as follows: Given the divided differences:*

$$x_{n,t}^{\text{diff}} := \frac{x_n(t) - x_n(t - \epsilon)}{\epsilon}, \quad t = \epsilon, 2\epsilon, \ldots, T$$

*Then the average of the point-wise std. of the divided differences converges to the MPGD:*

$$\frac{\epsilon}{T} \sum_{t=\epsilon}^{T} \widehat{\text{std}}[(x_{n,t}^{\text{diff}})_{n=1:N}] \longrightarrow \text{MPGD}(p) \quad \text{for} \quad \epsilon \to 0 \quad \text{and} \quad N \to \infty \quad \text{almost surely}$$ (9)

The proofs can be found in Appendix B. Due to Lemma 1 and Lemma 2, we can approximate the MGD, MPGD and therefore the JGD based on the set of evenly spaced sequences $X \in \mathbb{R}^{N \times T}$, where $x_{n,t}$ denotes the *value* at time-step $t$ in the $n$-th sequence, with $x_n \sim p$. Given $x_{n,t}^{\text{diff}} := x_{n,t+1} - x_{n,t}$, it holds that $\text{JGD}(p) \approx \text{MPGD}(p) \cdot \frac{1}{N} \sum_{n=1}^{N} \text{MGD}(x_n)$ and furthermore:

$$\text{MGD}(x_n) \approx \widehat{\text{std}}[x_n^{\text{diff}}] \qquad \text{MPGD}(p) \approx \frac{\epsilon}{T} \sum_{t=\epsilon}^{T} \widehat{\text{std}}[(x_{n,t}^{\text{diff}})_{n=1:N}]$$ (10)

Lemma 1 and Lemma 2 thus allow us to approximate the JGD-score by sampling the values of several functions on a shared grid from a generator. Its definition on the full function guarantees that we will always get the same results (in the bounds of the approximation error).

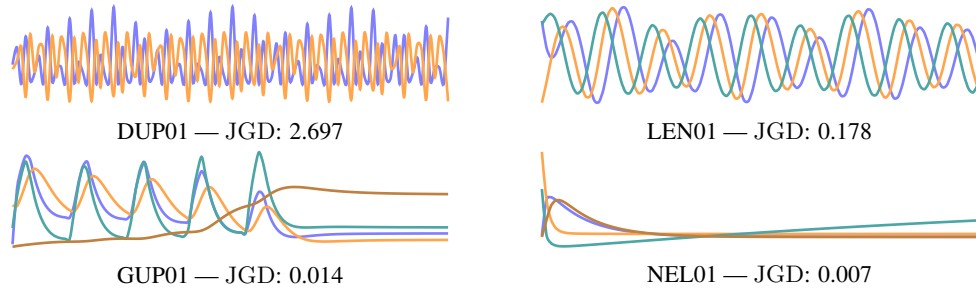

Figure 1: Demonstration of time series realized by 4 ODEs of different prediction difficulties without adding any noise. Each line / color represents a channel. Trajectories are shown for a duration of the respective $\sigma_{\text{dur}}$ as shown in Table 4.

**JGD for ODE models.**    To simplify the initial definition, we considered the JGD for a univariate function $x\colon [0, T] \to \mathbb{R}$, and we interpret a single channel within a multivariate ODE system as such a function. Computing the JGD for a multivariate ODE model requires combining the JGD values of each channel. We found that simply averaging all channels disadvantages models with a high number of channels. To address this, we compute the mean of the ten channels with the highest JGD. For models with ten or fewer channels, we revert to using the mean of all channels. Note that JGD is only a useful measure when applied to standardized data, otherwise the units are incomparable. Consequently, we normalize each channel to have a mean of 0 and standard deviation of 1 over all time steps and time series instances.

**Varying Single ODE Instances.**    From the scientific literature one often gets single ODE instances: a triplet $(F, a^{\text{LIT}}, x_0^{\text{LIT}})$ of a parametrized ODE system $F$, its parameters/constants $a^{\text{LIT}}$ and the initial conditions $x_0^{\text{LIT}}$, all three describing some specific experiment. To generate data by sampling only from such a single ODE instance will create functions that differ only in observation time points and channels and in the noise. These instances are too similar to each other, without noise there is only one function, and hence the MPGD of the distribution would be zero. We therefore propose to also carefully vary (i) the initial conditions $x_0$, (ii) the ODE constants $a$ and (iii) the total duration $T$ and of the ODE sampling process. We sample each from a distribution that yields the values given in the scientific literature on expectation and is controlled by a spread parameter $\sigma$ each:

$$x_0 \sim p_{\text{initial}}(\,\cdot \mid x_0^{\text{LIT}}, \sigma_{\text{initial}}) \qquad a \sim p_{\text{const}}(\,\cdot \mid a^{\text{LIT}}, \sigma_{\text{const}}) \qquad T \sim p_{\text{dur}}(\,\cdot \mid \sigma_{\text{dur}}) \tag{11}$$

Let $p$ be the two-stage function generator that first samples tuples $(x_0, a, T)$ from their respective distributions and then yields as function $x$ the solution to the ODE $F$ with initial values $x_0$, constants $a$ and running for a duration of $T$. Instead of blindly choosing the spreads $\sigma$, we optimize them w.r.t. the JGD-value of the generator:

$$(\sigma_{\text{initial}}^*, \sigma_{\text{const}}^*, \sigma_{\text{dur}}^*) \coloneqq \underset{\sigma_{\text{initial}}, \sigma_{\text{const}}, \sigma_{\text{dur}}}{\arg\max} \ \text{JGD}\big(p(\,\cdot \mid \sigma_{\text{initial}}, \sigma_{\text{const}}, \sigma_{\text{dur}})\big) \tag{12}$$

The JGD of each generator is estimated by 100 randomly sampled fully observed time series with a sequence length of 100. Since we are interested in the difficulty of the forecasting task, we compute the JGD on the final 50 steps of sequence only. We search $\sigma_{\text{initial}}$ from the range of $\{0.1, 0.3, 0.5\}$, $\sigma_{\text{const}}$ from $\{0.05, 0.1, 0.3\}$, and $\sigma_{\text{dur}}$ from $\{0.33, 1, 3.3, 10, 30\}$. The unit of $\sigma_{\text{dur}}$ varies for each ODE model, it is given by Physiome (Yu et al., 2011) and listed in Table 4 (appendix).

**Creating the `Physiome-ODE` Benchmark.**    In the Physiome database we identified $N = 208$ multivariate ODE models $(F^n, a^{\text{LIT},n}, x_0^{\text{LIT},n})_{n=1:N}$, that are defined by automatically generated Python code. We ranked all ODE-models with according to their optimized JGD-value (see Table 4) and selected the highest 50 datasets for `Physiome-ODE`. For each of the final datasets, we created a dataset with 2000 instances each. To further increase the range of initial states, we create datasets of lengths $\sigma_{\text{dur}}^*$, but with 200 observation steps. For each time series instance, we sample the initial time step $t_{\text{onset}} \sim \{t_i\}_{1 \leq i \leq 100}$ and then select the next 100 steps for each time series instance. Consequently, each time series instance has a duration of $\frac{1}{2}\sigma_{\text{dur}}^*$. While the time series instances are sampled regularly at first, we create IMTS instances by randomly dropping observations with a chance of 80%. For $p_{noise}$ we select to add $\epsilon \sim \mathcal{N}(0, 0.05)$. In Figure 1 we show four trajectories contained in `Physiome-ODE` with the respective JGD.

**Handling exploding ODEs.** Varying the constants and initial states of the ODEs can lead to inconsistent states, which could never occur in real world scenarios. In general, this is not an issue regarding the machine learning perspective on our experiments. Occasionally however, certain combinations of inconsistent states and constants can lead to extreme values which are magnitudes greater than values that occur in other time series instances. These extreme values will dominate the training and evaluation loss in a problematic manner. We prevent this by rejecting all $(\sigma_{\text{initial}}, \sigma_{\text{const}}, \sigma_{\text{dur}})$ containing any values greater than 10 times the channel-wise standard deviation in the 100 samples created for evaluation. If such values nevertheless occur within the final 2000 samples created for Physiome-ODE, we drop the respective instances. We follow the same protocol to handle errors thrown by the ode-solver.

## 5 EXPERIMENTS

**Baseline models.** We compare on 5 recent IMTS forecasting models, namely **GRU-ODE-Bayes** [1] (De Brouwer et al., 2019), **Neural Flows** (Biloš et al., 2021), Continuous Recurrent Unit (**CRU**; Schirmer et al., 2022), **LinODEnet** (Scholz et al., 2022; Mione et al., 2024) and **GraFITi** (Yalavarthi et al., 2024). GRU-ODE-Bayes, Neural Flows, CRU and LinODEnet are differential equations based models whereas GraFITi is a graph based model. We also introduce a naive model called GraFITi-Constant (GraFITi-C) which predicts a fixed value for all the future time points in all the target channels for an instance.

**Experimental Protocol.** In our experiments models have to predict the last 50% of the time series after observing the first 50%. We use 5-fold cross validation, and for each fold we split the data into training, validation and test set with a ratio of 70:20:10. Additionally, we resample the sparse observation mask to transform each instance into an IMTS. The evaluation metric is the mean square error (MSE). For simplicity, we randomly sample 10 hyperparameter configurations and fitted each model on a single fold per dataset, selecting the configuration with the lowest validation MSE. The winning configuration per model is then trained and evaluated on all 5 folds (see Appendix F). We run the benchmark experiments on a cluster of 4 NVIDIA 4090 GPUs with 24 GB.

**Empirical Results.** The numbers presented in Table 2 are the mean and standard deviation of MSE for 5 folds. Overall, LinODEnet (Scholz et al., 2022; Mione et al., 2024) is the best-performing model, winning the most datasets and having the highest average rank. On the other hand, the average rank of GraFITi (2.20) is close to that of LinODEnet (2.10). Surprisingly, GraFITi-C performs better than Neural Flows and GRU-ODE-Bayes and has an average rank of 2.86. This is because datasets like DUP01 and DOK01 are too hard to forecast, hence the complex models cannot improve upon the accuracy of a constant model. Other datasets are too simple that even a constant model yields MSE close to 0.0025, which is the lowest value possible to achieve due to the normal-distributed noise with a variance of 0.05.

In order to check how good our JGD score can measure the difficulty in forecasting of a time series we plot JGD score vs the best MSE score achieved in Figure 2. It can be observed that JGD fulfills its purpose of finding challenging and interesting ODE models. More specifically JGD and the test MSE of the best performing model are correlated with a Spearman coefficient of 0.77.

## 6 EXTENSIONS AND LIMITATIONS

Currently, most datasets in Physiome-ODE still show a good potential for improvement: even the best performing methods are far away from the Bayes error of 0.0025 MSE. In case authors of future methods get the impression that their method could benefit from larger datasets, i.e., more instances per dataset than 2000, they easily can generate larger ones with 10,000, 100,000, etc. instances.

Physiome-ODE is not based on observation data as in these areas observed data usually is expensive to get, and thus only limited amounts exists or are publically available, at least. In fact, we scanned through the respective papers of our top 50 ODE models and could not find the original experimental data for a single one. Hence, if one wants to conduct machine learning experiments on these complex and rich biological processes one is forced to create them using the ODE models describing the process.

---

[1]For brevity, we denote this as GRU-ODE in the tables.

Table 2: Experimental results on various baseline models. `Physiome-ODE` datasets ranked by JGD-score. We highlight the best model in **bold** and underline the second best.

| Dataset | JGD | GRU-ODE | LinODEnet | CRU | Neural Flows | GraFITi | GraFITi-C |
|---------|-----|---------|-----------|-----|--------------|---------|-----------|
| DUP01 | 2.697 | 1.037± 0.047 | 0.964± 0.036 | 0.958± 0.038 | 1.030± 0.046 | 0.955± 0.037 | **0.951± 0.036** |
| JEL01 | 2.580 | 1.017± 0.014 | 0.949± 0.015 | 0.939± 0.015 | 1.000± 0.013 | 0.942± 0.020 | **0.935± 0.016** |
| DOK01 | 2.277 | 1.011± 0.004 | 0.996± 0.003 | 0.985± 0.005 | 0.998± 0.005 | 0.984± 0.005 | **0.982± 0.005** |
| INA01 | 2.218 | 1.018± 0.010 | 1.009± 0.011 | 1.005± 0.010 | 1.008± 0.008 | **1.004± 0.010** | **1.004± 0.009** |
| WOL01 | 1.973 | 0.952± 0.035 | 0.806± 0.027 | 0.814± 0.029 | 0.841± 0.029 | 0.787± 0.028 | **0.784± 0.030** |
| BOR01 | 1.795 | 0.794± 0.034 | 0.719± 0.021 | 0.715± 0.020 | 0.743± 0.026 | 0.712± 0.022 | **0.709± 0.022** |
| HYN01 | 1.548 | 0.883± 0.060 | 0.672± 0.044 | 0.665± 0.053 | 0.636± 0.045 | 0.625± 0.043 | **0.619± 0.046** |
| JEL02 | 1.271 | 0.816± 0.049 | 0.693± 0.031 | **0.674± 0.028** | 0.779± 0.028 | 0.699± 0.029 | 0.687± 0.027 |
| DUP02 | 1.202 | 0.895± 0.055 | 0.740± 0.042 | 0.722± 0.046 | 0.890± 0.081 | 0.728± 0.044 | **0.718± 0.046** |
| WOL02 | 0.895 | 0.854± 0.010 | 0.663± 0.015 | 0.653± 0.017 | 0.685± 0.012 | 0.654± 0.014 | **0.645± 0.016** |
| DIF01 | 0.735 | 1.035± 0.023 | **0.832± 0.087** | 0.985± 0.025 | 1.014± 0.025 | 0.985± 0.030 | 0.982± 0.029 |
| VAN01 | 0.407 | 0.321± 0.023 | 0.250± 0.006 | 0.253± 0.005 | 0.250± 0.006 | 0.246± 0.005 | **0.242± 0.006** |
| DUP03 | 0.254 | 0.874± 0.089 | 0.632± 0.044 | **0.622± 0.047** | 1.098± 0.447 | 0.627± 0.043 | 0.744± 0.042 |
| BER01 | 0.179 | 0.594± 0.054 | **0.279± 0.020** | 0.280± 0.016 | 0.398± 0.014 | 0.300± 0.018 | 0.342± 0.018 |
| LEN01 | 0.178 | 1.028± 0.060 | **0.387± 0.071** | 0.754± 0.157 | 1.009± 0.061 | 0.607± 0.055 | 0.970± 0.063 |
| LI01 | 0.113 | 1.067± 0.015 | **0.084± 0.009** | 0.175± 0.020 | 0.979± 0.049 | 0.202± 0.013 | 0.742± 0.010 |
| LI02 | 0.097 | 0.723± 0.080 | 0.434± 0.044 | 0.437± 0.046 | 0.674± 0.105 | **0.397± 0.058** | 0.458± 0.056 |
| REV01 | 0.081 | 1.091± 0.075 | **0.597± 0.061** | 0.602± 0.049 | 0.978± 0.042 | 0.674± 0.055 | 0.855± 0.050 |
| PUR01 | 0.049 | 0.703± 0.058 | **0.106± 0.006** | 0.353± 0.083 | 0.654± 0.102 | 0.153± 0.006 | 0.476± 0.020 |
| NYG01 | 0.047 | 0.571± 0.074 | 0.358± 0.071 | 0.403± 0.092 | 0.442± 0.094 | **0.344± 0.065** | 0.366± 0.047 |
| PUR02 | 0.044 | 0.830± 0.066 | **0.280± 0.028** | 0.293± 0.026 | 0.723± 0.077 | 0.322± 0.021 | 0.511± 0.023 |
| HOD01 | 0.042 | 0.851± 0.118 | 0.441± 0.043 | **0.409± 0.049** | 0.701± 0.077 | 0.493± 0.046 | 0.609± 0.056 |
| REE01 | 0.035 | 0.266± 0.068 | 0.045± 0.012 | 0.051± 0.008 | 0.045± 0.011 | **0.033± 0.007** | 0.039± 0.012 |
| VIL01 | 0.028 | 0.511± 0.053 | 0.374± 0.021 | 0.373± 0.039 | 0.500± 0.060 | **0.344± 0.044** | 0.378± 0.042 |
| KAR01 | 0.023 | 0.193± 0.013 | **0.034± 0.008** | 0.044± 0.012 | 0.069± 0.009 | 0.041± 0.013 | 0.078± 0.011 |
| SHO01 | 0.023 | 0.260± 0.017 | 0.057± 0.006 | 0.095± 0.010 | 0.092± 0.014 | 0.062± 0.013 | **0.055± 0.013** |
| BUT01 | 0.020 | 0.583± 0.172 | **0.254± 0.074** | 0.317± 0.108 | 0.441± 0.153 | 0.281± 0.071 | 0.324± 0.091 |
| MAL01 | 0.019 | 0.420± 0.061 | **0.018± 0.007** | 0.064± 0.007 | 0.052± 0.002 | 0.020± 0.004 | 0.054± 0.005 |
| ASL01 | 0.019 | 0.114± 0.035 | **0.022± 0.003** | 0.046± 0.014 | 0.066± 0.033 | 0.025± 0.009 | 0.026± 0.002 |
| BUT02 | 0.016 | 0.483± 0.047 | **0.207± 0.056** | 0.282± 0.042 | 0.329± 0.068 | 0.248± 0.052 | 0.256± 0.039 |
| MIT01 | 0.015 | 0.005± 0.001 | **0.003± 0.000** | **0.003± 0.000** | 0.008± 0.004 | **0.003± 0.000** | **0.003± 0.000** |
| GUP01 | 0.014 | 0.153± 0.041 | **0.018± 0.007** | 0.057± 0.017 | 0.043± 0.017 | 0.041± 0.006 | 0.035± 0.006 |
| GUY01 | 0.013 | 0.036± 0.004 | 0.006± 0.005 | **0.004± 0.001** | 0.024± 0.015 | 0.005± 0.003 | **0.004± 0.001** |
| PHI01 | 0.013 | 0.674± 0.030 | **0.131± 0.014** | 0.133± 0.020 | 0.635± 0.158 | 0.222± 0.013 | 0.345± 0.015 |
| GUY02 | 0.013 | 0.124± 0.044 | **0.010± 0.006** | **0.010± 0.002** | 0.082± 0.066 | 0.012± 0.009 | 0.032± 0.015 |
| PUL01 | 0.013 | 0.099± 0.024 | **0.008± 0.004** | 0.012± 0.003 | 0.061± 0.060 | **0.008± 0.001** | 0.024± 0.008 |
| CAL01 | 0.013 | 1.049± 0.055 | **0.078± 0.009** | 0.158± 0.008 | 0.867± 0.014 | 0.179± 0.012 | 0.643± 0.024 |
| WOD01 | 0.012 | 0.612± 0.072 | 0.154± 0.016 | **0.113± 0.017** | 0.510± 0.060 | 0.164± 0.013 | 0.344± 0.016 |
| GUP02 | 0.012 | 0.870± 0.059 | 0.469± 0.022 | **0.444± 0.018** | 0.577± 0.017 | 0.449± 0.027 | 0.461± 0.025 |
| M01 | 0.012 | 0.055± 0.009 | 0.004± 0.001 | 0.005± 0.000 | 0.124± 0.207 | **0.003± 0.000** | **0.003± 0.000** |
| LEN02 | 0.012 | 0.297± 0.071 | **0.039± 0.005** | 0.059± 0.012 | 0.380± 0.141 | 0.099± 0.021 | 0.143± 0.022 |
| KAR02 | 0.011 | 0.515± 0.032 | **0.140± 0.010** | 0.151± 0.011 | 0.257± 0.016 | 0.151± 0.009 | 0.252± 0.010 |
| SHO02 | 0.011 | 0.368± 0.028 | **0.037± 0.006** | 0.083± 0.015 | 0.109± 0.015 | 0.043± 0.006 | 0.073± 0.010 |
| MAC01 | 0.010 | 0.242± 0.026 | 0.020± 0.003 | 0.065± 0.006 | 0.029± 0.011 | 0.021± 0.003 | **0.019± 0.002** |
| IRI01 | 0.010 | 0.151± 0.032 | **0.037± 0.003** | 0.049± 0.010 | 0.116± 0.006 | 0.038± 0.017 | 0.097± 0.008 |
| BAG01 | 0.010 | 0.294± 0.041 | 0.032± 0.005 | 0.046± 0.005 | 0.075± 0.012 | **0.029± 0.002** | 0.109± 0.002 |
| WOL03 | 0.008 | 0.859± 0.114 | **0.073± 0.010** | 0.177± 0.016 | 0.479± 0.107 | 0.105± 0.016 | 0.247± 0.032 |
| WAN01 | 0.008 | 0.504± 0.031 | **0.103± 0.012** | 0.125± 0.012 | 0.345± 0.027 | 0.119± 0.010 | 0.232± 0.015 |
| NEL01 | 0.007 | 0.054± 0.012 | 0.010± 0.001 | 0.009± 0.001 | 0.060± 0.029 | **0.007± 0.000** | 0.023± 0.006 |
| HUA01 | 0.007 | 0.321± 0.046 | **0.052± 0.004** | 0.116± 0.007 | 0.149± 0.015 | 0.063± 0.005 | 0.115± 0.007 |
| # Wins | — | 0 | **25** | 8 | 2 | 10 | 15 |
| Rank | — | 5.9 | **2.10** | 2.82 | 4.82 | 2.20 | 2.86 |

Consequently, we build the benchmark on top of real-world ODEs: their system, their constants and their initial values establish a real-world connection that has been created in hundreds of scientific publications. To delineate our benchmark from purely synthetic data we therefore call it *semi-synthetic*.

The fact that the datasets are generated by ODEs implies that some general ODE-based models are expressive enough to mimic the generating process and to perform well for large amounts of data. For example, LinODEnet can represent a Koopman-type (Koopman, 1931; Koopman and v. Neumann, 1932) linear approximation of any nonlinear ODE under some assumptions. However, being expressive enough does not mean that these models are best when trained on a limited amount of data. Other, non-ODE models can beat ODE-based models with

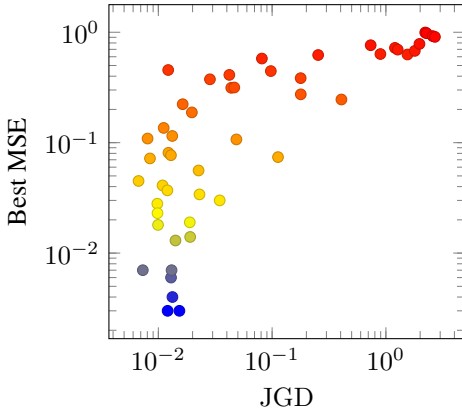

Figure 2: Test MSE of the best performing model vs JGD-score across 50 datasets.

limited training data, even if the ground truth is an ODE. In fact, a main purpose of the benchmark is to answer questions about the best forecasting models for data generated from an (unknown) ODE ground truth. Despite our limitation to ODEs from biology the perspective is forecasting for many scientific domains where dynamics are *in principle* governed by ODEs.

The inductive bias of ODE-based models (LinODEnet (Scholz et al., 2022)) is rather weak. Being based on ODEs does not mean that these models would only perform well on data generated from ODEs. When one spells out the expressiveness of LinODEnet in terms of basis functions, it would just mean that after a nonlinear transformation the time dependent functions can be expressed by some basis functions given as solutions of some linear ODE. The representation is entirely trainable and nothing about the time series in question is hard-coded apart from smoothness.

The number of instances per dataset contained in `Physiome-ODE` is deliberately small (2000). Our goal was not to create huge datasets just for the sake of being huge, but to create datasets for the purpose of research: the smallest datasets that would allow complex, ODE-based models to demonstrate their strengths. However, it is trivial to create an arbitrary number of time series, if larger datasets are desired.

While this work focuses on IMTS forecasting, `Physiome-ODE` can be easily extended to regular time series forecasting. In Appendix G, we conduct experiments with three datasets created with `Physiome-ODE`. Our results show that these datasets have an interesting property: They force models to learn channel dependencies. This contradicts with the results on traditional evaluation datasets, in which models like PatchTST (Nie et al., 2023) are more effective when they ignore inter-channel correlations.

# 7 PRIOR TIME SERIES FORECASTING BENCHMARKS

**Monash time series forecasting archive (Godahewa et al., 2021)** The Monash repository is the first benchmark for time series forecasting. It encompasses 11 openly-accessible multivariate time series datasets drawn from various domains, including energy, banking, and nature. Some of these datasets may include missing values. However, the sampling process is regular. Therefore, the missing values can easily be imputed, so that methods designed for regular time series forecasting can be applied. Hypothetically, we could use the 11 multivariate time series contained in the Monash archive to create IMTS datasets for an IMTS forecasting benchmark. However, most of these multivariate datasets are actually multiple univariate time series stacked on top of each other and/or a single long time series that is segmented to create multiple time series instances. Other datasets used to evaluate regular multivariate time series forecasting (Nie et al., 2023; Zeng et al., 2023; Zhou et al., 2021), are not well-suited for IMTS forecasting for similar reasons. In fact, Nie et al. (2023) showed that models benefit, when they ignore channel correlations, hinting that these are barely carrying

useful information in the respective datasets. In IMTS forecasting however, modeling these channel correlations is a crucial property of the models, due to the high number of missing values.

**Chaotic ODEs benchmark for time series forecasting**   Closely related to `Physiome-ODE`, Gilpin (2021) provides a forecasting benchmark created with chaotic ordinary differential equations that mainly originate from Physics. Similar to the Monash repository, each chaotic ODE creates single long trajectory, without varying the constants. In contrast, `Physiome-ODE` contains 2000 different trajectories with varying constants and initial states. Furthermore, the inherent chaotic nature of these equations often presents a forecasting challenge, even in the regularly sampled and fully observed forecasting setting. On the other hand, IMTS forecasting is challenging due to the sparse observation of a time series. We assume that combining these two factors of difficulty, would surpass any current methods forecast modeling ability. This is closely related to what we observe on the most difficult dataset of `Physiome-ODE`. We support our assumption, by an example experiment on the Lorenz-attractor, a famous chaotic ODE, which is a part of Gilpin (2021)'s benchmark. Respective experimental details are given in Appendix H and results in Table 7. Similarly, to certain `Physiome-ODE` datasets with a high JGD (e.g. DUP01), no method can significantly outperform the simple baseline GraFITi-C.

**PDEBench (Takamoto et al., 2022)**   Related to applying differential equations to create datasets, there exists a benchmark with 11 spatio-temporal datasets named PDEBench. Here, models are trained to estimate the parameters of an underlying partial differential equation (PDE) used to generate such dataset, instead of forecasting. Current IMTS forecasting architectures cannot utilize the spatio-temporal information inherent in PDEBench's datasets.

## 8 CONCLUSION

We introduced `Physiome-ODE`, the first ever IMTS forecasting benchmark. `Physiome-ODE`'s datasets are created with ODE models, that were defined in decades of research and published on the Physiome Model Repository. Our experiments showed that LinODEnet and CRU are actually better than previous evaluation on established datasets indicated. Nevertheless, we also provided a few datasets, on which models are unable to outperform a constant baseline model. We believe that our datasets, especially the very difficult ones, can help to identify deficits of current architectures and support future research on IMTS forecasting.

## REPRODUCIBILITY STATEMENT

Our experiments can be reproduced by following the instructions provided in our Git repository: `https://github.com/kloetergensc/Physiome-ODE`. There are two options to obtain the datasets from `Physiome-ODE`. The first is to regenerate everything using the code and instructions in the repository. The second is to download the data from Zenodo `https://zenodo.org/records/11492058`:

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

## A  REAL WORLD DATASET STATISTICS

Table 3: Statistics of evaluation datasets used by De Brouwer et al. (2019); Biloš et al. (2021); Schirmer et al. (2022). *Max. Len.* refers to the maximum sequence length among samples. *Max. Obs.* refers to the maximum number of non-missing observations among samples. *Sparsity* refers to the percentage of missing values over all samples.

| name | Instances | Channel | Max. Len | Max. Obs | Spars. |
|------|-----------|---------|----------|----------|--------|
| USHCN | 1.114 | 5 | 370 | 398 | 78.0% |
| PhysioNet-2012 | 11.981 | 37 | 48 | 606 | 80.4% |
| MIMIC-III | 21.250 | 96 | 97 | 677 | 94.2% |
| MIMIC-IV | 17.874 | 102 | 920 | 1642 | 97.8% |

## B  PROOFS

### B.1  PROOF OF LEMMA 1

*Proof.* Due to Equation (5), it is sufficient to show that

$$\widehat{\text{std}}[x^{\text{diff},\epsilon}]^2 \xrightarrow{\epsilon \to 0} \frac{1}{T} \int_{t=0}^{T} \left( x'(t) - \frac{(x(T) - x(0))}{T} \right)^2 \mathrm{d}t. \tag{13}$$

Due to the mean value theorem, there exists $\zeta_{t,\epsilon} \in [t - \epsilon, t]$ with $\frac{x(t) - x(t-\epsilon)}{\epsilon} = x'(\zeta_{t,\epsilon})$ for all $t = \epsilon, 2\epsilon, \ldots, T$. Given $N := \frac{T}{\epsilon}$ and $\mu_N := \frac{1}{N} \sum_{t=\epsilon,2\epsilon,\ldots,T} x_t^{\text{diff},\epsilon}$, we thus have

$$\mu_N = \frac{1}{N} \sum_{t=\epsilon,2\epsilon,\ldots,T} x'(\zeta_{t,\epsilon}) \tag{14a}$$

$$= \frac{1}{T} \sum_{t=\epsilon,2\epsilon,\ldots,T} \underbrace{\frac{T}{N}}_{=\epsilon} x'(\zeta_{t,\epsilon}) \tag{14b}$$

$$\xrightarrow{\epsilon \to 0} \frac{1}{T} \int_{t=0}^{T} x'(t)\,\mathrm{d}t \tag{14c}$$

$$= \frac{x(T) - x(0)}{T} \tag{14d}$$

due to the definition of the Riemann integral. Furthermore, we have

$$\widehat{\text{std}}[x^{\text{diff},\epsilon}]^2 = \frac{1}{N} \left( \sum_{t=\epsilon,2\epsilon,\ldots,T} (x_t^{\text{diff},\epsilon})^2 \right) - \mu_N^2 \tag{15}$$

Therefore, again by the mean value theorem and the definition of the Riemann integral:

$$\frac{1}{N} \left( \sum_{t=\epsilon,2\epsilon,\ldots,T} (x_t^{\text{diff},\epsilon})^2 \right) = \frac{1}{N} \left( \sum_{t=\epsilon,2\epsilon,\ldots,T} x'(\zeta_{t,\epsilon})^2 \right) \tag{16a}$$

$$= \frac{1}{T} \sum_{t=\epsilon,2\epsilon,\ldots,T} \underbrace{\frac{T}{N}}_{=\epsilon} x'(\zeta_{t,\epsilon})^2 \tag{16b}$$

$$\xrightarrow{\epsilon \to 0} \frac{1}{T} \int_{t=0}^{T} x'(t)^2\,\mathrm{d}t \tag{16c}$$

Consequently,

$$\widehat{\text{std}}[x^{\text{diff},\epsilon}]^2 \xrightarrow{\epsilon \to 0} \frac{1}{T} \int_{t=0}^{T} x'(t)^2\,\mathrm{d}t - \left( \frac{(x(T) - x(0))}{T} \right)^2 \tag{17}$$

$$= \frac{1}{T} \int_{t=0}^{T} \left( x'(t) - \frac{(x(T) - x(0))}{T} \right)^2 \mathrm{d}t. \tag{18}$$

$$\square$$

## B.2 ASSUMPTIONS AND PROOF OF LEMMA 2

For the lemma to hold we make the technical assumption:

(A) The distribution $p$ of continuous functions $[0, T] \rightarrow \mathbb{R}$ is given by a continuous map[2] $\phi \colon \Theta \times [-1, T] \rightarrow \mathbb{R}, (\theta, t) \mapsto \phi(\theta, t)$, with a compact parameter space $\Theta \subset \mathbb{R}^d$, and a probability measure $\mu$ on $\Theta$. We require that the map $\frac{\partial \phi}{\partial t}$ is continuous, too.

We sometimes use $x_\theta$ for $\phi(\theta, \cdot)$ and $x'_\theta$ for $\frac{\partial \phi}{\partial t}(\theta, \cdot)$. This is no restriction for the differential equations we consider, as we can always obtain a solution on $[-1, T]$, given parameters (including $x(0) = x_0$).

Sampling from this distribution is done by sampling from the probability measure on $\Theta$. A sequence of i.i.d. random variables $\theta_n$, $n \in \mathbb{N}_{>0}$, distributed according to $\mu$, defines a sequence of random functions $(x_n)_{n \in \mathbb{N}_{>0}}$, $x_n(t) = \phi(\theta_n, t)$, as in the statement of the Lemma 2.

For the proof we will need a uniform law of large numbers (ULLN). There exist many versions of such theorems (e.g. (Jennrich, 1969), or (Andrews, 1987)), but we prefer to state and prove a version with strong assumptions appropriate for our setting.

**Lemma 3** (Uniform Law of Large Numbers). *Let $f \colon \Theta \times I \rightarrow \mathbb{R}$ be a continuous function on compact metric spaces $\Theta$ and $I$. Let $\theta_i$, $i \in \mathbb{N}_{>0}$ be i.i.d. random variables with values in $\Theta$, all distributed according to a probability measure $\mu$ on $\Theta$. Then*

$$\frac{1}{N} \sum_{i=1}^{N} f(\theta_i, t) \xrightarrow{N \rightarrow \infty} \int_{\Theta} f(\theta, t) \, \mathrm{d}\mu(\theta) \tag{19}$$

*uniformly in $t \in I$ almost surely.*

*Proof.* As $\Theta \times I$ is compact and $f$ continuous, $f$ is even uniformly continuous. Hence, for any $\epsilon > 0$, we can then choose a $\delta$ such that

$$d(\theta, \theta') < \delta \quad \text{and} \quad (t, t') < \delta \Longrightarrow |f(\theta', t') - f(\theta, t)| < \epsilon \tag{20}$$

Where $d$ is the metric on $\Theta$. Now by compactness of $I$, finitely many $\delta$-balls in $I$ cover it, i.e. there are $t_1, \ldots, t_k \in I$, such that the $B_\delta(t_j)$, $j = 1, \ldots, k$ cover $I$. For each $(t_j)_{j=1:k}$ we can apply a strong law of large numbers to obtain almost surely convergence of $\frac{1}{N} \sum_{i=1}^{N} f(\theta_i, t_j)$ to $\int_{\Theta} f(\theta, t_j) \, \mathrm{d}\mu(\theta)$.

Hence, almost surely, the following holds: For $\epsilon > 0$ we can choose $N_0 > 0$ such that for all natural numbers $N \geq N_0$ and all $j \in \{1, \ldots, k\}$:

$$\left| \frac{1}{N} \sum_{i=1}^{N} f(\theta_i, t_j) - \int_{\Theta} f(\theta, t_j) \, \mathrm{d}\mu(\theta) \right| < \epsilon \tag{21}$$

Now for any $t \in I$ there is a $j \in \{1, \ldots, k\}$ such that

$$\left| \frac{1}{N} \sum_{i=1}^{N} \big( f(\theta_i, t_j) - f(\theta_i, t) \big) \right| < \epsilon \qquad \text{and} \qquad \left| \int_{\Theta} f(\theta, t_j) - f(\theta, t) \, \mathrm{d}\mu(\theta) \right| < \epsilon \tag{22}$$

The triangle inequality yields

$$\left| \frac{1}{N} \sum_{i=1}^{N} f(\theta_i, t) - \int_{\Theta} f(\theta, t) \, \mathrm{d}\mu(\theta) \right| < 3\epsilon \tag{23}$$

for all $t \in I$, which proves the claim. $\square$

*Proof.* (Lemma 2) Let $\theta_i$, $i \in \mathbb{N}_{>0}$ be i.i.d. random variables with values in $\Theta$, all distributed according to a probability measure $\mu$ on $\Theta$.

---

[2]We define $\phi$ on the larger interval $[-1, T]$, instead of $[0, T]$, in order to use one-sided difference quotients.

Define $\psi \colon \Theta \times [0,1] \times [0,T]$ by

$$\psi(\theta, \epsilon, t) = \begin{cases} \frac{\phi(\theta, t) - \phi(\theta, t - \epsilon)}{\epsilon} & \text{if } \epsilon > 0 \\ \frac{\partial \phi}{\partial t}(\theta, t) & \text{if } \epsilon = 0 \end{cases} \tag{24}$$

For $\epsilon > 0$ the function is obviously continuous as a composition of continuous functions. Let now be $\epsilon = 0$ and $t \in [0,T]$, $\theta \in \Theta$. We claim that $\psi$ is continuous at $(\theta, 0, t)$. $\frac{\partial \phi}{\partial t}$ is uniformly continuous on its compact domain of definition.

Let $\delta$ be greater than zero. We can choose a $\gamma > 0$ such that for $d(\theta, \theta') < \gamma$ and $|t - t'| < \gamma$ we have $\left| \frac{\partial \phi}{\partial t}(\theta', t') - \frac{\partial \phi}{\partial t}(\theta, t) \right| < \delta$.

Now chose $(\theta', \epsilon', t')$ with $d(\theta, \theta') < \gamma$ and $d(t, t') < \gamma/2$ and $0 \le \epsilon' < \gamma/2$. If $\epsilon' = 0$, we directly obtain

$$|\psi(\theta', 0, t') - \psi(\theta, 0, t)| = \left| \frac{\partial \phi}{\partial t}(\theta', t') - \frac{\partial \phi}{\partial t}(\theta, t) \right| < \delta \tag{25}$$

If $\epsilon' > 0$, the mean value theorem implied that there is a $\tau \in (t' - \epsilon', t')$ such that:

$$|\psi(\theta', \epsilon', t') - \psi(\theta, 0, t)| = \left| \frac{\partial \phi}{\partial t}(\theta', \tau) - \frac{\partial \phi}{\partial t}(\theta, t) \right| < \delta \tag{26}$$

as $d(\tau, t) \le d(\tau, t') + d(t', t) \le \epsilon' + \gamma/2 < \gamma/2 + \gamma/2 = \gamma$. This establishes the continuity of $\psi$.

Therefore, $\psi$ is also uniformly continuous on its compact domain of definition.

Define $\Psi \colon [0,1] \times [0,T] \to \mathbb{R}$ by

$$\Psi(\epsilon, t) = \int_\Theta \psi(\theta, \epsilon, t) \, d\mu(\theta) \tag{27}$$

We also define $\Psi^N \colon [0,1] \times [0,T] \to \mathbb{R}$

$$\Psi^N(\epsilon, t) = \frac{1}{N} \sum_{i=1}^N \psi(\theta_i, \epsilon, t) \tag{28}$$

The uniform law of large numbers shows that almost surely $\Psi^N$ converges uniformly to $\Psi$ for $N \to \infty$.

Analogously we define $\Xi \colon [0,1] \to \mathbb{R}$ by

$$\Xi(\epsilon, t) = \int_\Theta \psi(\theta, \epsilon, t)^2 \, d\mu(\theta) \tag{29}$$

We also define $\Psi^N \colon [0,1] \times [0,T] \to \mathbb{R}$

$$\Xi^N(\epsilon, t) = \frac{1}{N} \sum_{i=1}^N \psi(\theta_i, \epsilon, t)^2 \tag{30}$$

Again, the uniform law of large numbers shows that almost surely $\Xi^N$ converges uniformly to $\Xi$ for $N \to \infty$. Hence,

$$\widehat{\text{std}}[(\psi(\theta_i, \epsilon, t))_{i=(1:N)}] = \sqrt{\Xi^N(\epsilon, t) - \Psi^N(\epsilon, t)^2} \tag{31}$$

converges almost surely uniformly to $\sqrt{\Xi(\epsilon, t) - \Psi(\epsilon, t)^2} = \text{std}_{\theta \sim \mu}[\Psi(\epsilon, t)]$ for $N \to \infty$.

On the other hand by uniform continuity

$$\widehat{\text{std}}[(\psi(\theta_i, \epsilon, t))_{i=(1:N)}] \xrightarrow[\text{uniformly}]{\epsilon \to 0} \widehat{\text{std}}[(\psi(\theta_i, 0, t))_{i=(1:N)}] \tag{32}$$

$$\sqrt{\Xi(\epsilon, t) - \Psi(\epsilon, t)^2} \xrightarrow[\text{uniformly}]{\epsilon \to 0} \text{std}_{\theta \sim \mu}[(\Psi(0, t))] = \text{std}_{x \sim p} x'(t) \tag{33}$$

Now, given some $\delta > 0$, we can choose an $\epsilon'$ and almost surely an $N_0$ such that for all $0 < \epsilon < \epsilon'$ and $N \geq N_0$:

$$\left| \widehat{\text{std}}[(\psi(\theta_i, \epsilon, t))_{i=(1:N)}] - \widehat{\text{std}}[(\psi(\theta_i, 0, t))_{i=(1:N)}] \right|_{C^0([0,T])} \leq \delta \tag{34}$$

$$\left| \widehat{\text{std}}[(\psi(\theta_i, 0, t))_{i=(1:N)}] - \text{std}_{x \sim p} \, x'(t) \right|_{C^0([0,T])} \leq \delta \tag{35}$$

and for $t, t' \in [0, T]$, $|t - t'| < \epsilon$:

$$\left| \widehat{\text{std}}[(\psi(\theta_i, \epsilon, t))_{i=(1:N)}] - \widehat{\text{std}}[(\psi(\theta_i, \epsilon, t'))_{i=(1:N)}] \right| \leq \delta \tag{36}$$

Therefore,

$$\left| \frac{1}{T} \int_0^T \widehat{\text{std}}[(\psi(\theta_i, \epsilon, t))_{i=(1:N)}] \, \mathrm{d}t - \frac{1}{T} \int_0^T \text{std}_{x \sim p} \, x'(t) \, \mathrm{d}t \right| < 2\delta \tag{37}$$

and for $T/\epsilon \in \mathbb{N}$ the Riemann sum with step size $\epsilon$ differs from the integral by at most $T\delta$, i.e.

$$\left| \frac{1}{T} \int_0^T \widehat{\text{std}}[(\psi(\theta_i, \epsilon, t))_{i=(1:N)}] \, \mathrm{d}t - \frac{\epsilon}{T} \sum_{k=1}^{T/\epsilon} \widehat{\text{std}}[(\psi(\theta_i, \epsilon, k\epsilon))_{i=(1:N)}] \right| < \delta \tag{38}$$

In the notations of the lemma, with $\frac{1}{T} \int_0^T \text{std}_{x \sim p} \, x'(t) \, \mathrm{d}t = \text{MPGD}(p)$ we get:

$$\left| \frac{\epsilon}{T} \sum_{t=\epsilon}^{T} \widehat{\text{std}}[(x_{i,t}^{\text{diff}})_{i=1:N}] - \text{MPGD}(p) \right| < 3\delta \tag{39}$$

Thus proving the claim that the left term of the difference converges almost surely for $\epsilon \to 0$ and $N \to \infty$ to the right term of the difference. Please note that $N$ and $\epsilon$ were chosen independently. The limits can be taken simultaneously or in arbitrary order. $\qquad \square$

## C  RELATION OF MGD AND LIPSCHITZ CONSTANT

Regarding the relationship between the MGD and the Lipschitz constant of the vector field, we can consider the standard example of a linear ODE: $\dot{x}(t) = a \cdot x(t)$, for which the Lipschitz constant is precisely $a$. For simplicity, assume the initial condition $x(0) = 1$. Hence, the solution is $x(t) = e^{at}$. Then, by (4) we have $c = \frac{1}{T}(e^{aT} - 1)$, and hence:

$$\frac{1}{T} \int_0^T \|\dot{x}(t) - c\|^2 \, dt = \left(\frac{a}{2T} - \frac{1}{T^2}\right) e^{2aT} + \frac{1}{2T^2} e^{aT} - \frac{a}{2T} - \frac{1}{T^2}$$

$$= \mathcal{O}\left(\frac{a}{T} e^{2aT}\right)$$

$$\implies \text{MGD} = \mathcal{O}\left(\sqrt{\frac{a}{T}} e^{aT}\right)$$

In particular, for this example, the MGD grows super-exponentially with the Lipschitz constant. This result can be extended to an upper bound for general systems:

1. By the Mean Value Theorem, there exists $\xi \in (0, T)$ such that $c = \frac{x(T) - x(0)}{T} = \dot{x}(\xi) = f(x(\xi))$.

2. Therefore, $\|\dot{x}(t) - c\|^2 = \|\dot{x}(t) - \dot{x}(\xi)\|^2 = \|f(x(t)) - f(x(\xi))\|^2 \le L^2 \|x(t) - x(\xi)\|^2$.

3. We can bound the latter quantity via Grönwall's Lemma:

$$\|x(t) - x(\xi)\| = \left\|\int_\xi^t \dot{x}(s) \, ds\right\|$$

$$= \left\|\int_\xi^t f(x(s)) \, ds\right\|$$

$$\le \int_\xi^t \|f(x(s))\| \, ds \quad \text{assuming } t \ge \xi$$

$$\le \int_\xi^t \|f(x(s)) - f(x(\xi))\| + \|f(x(\xi))\| \, ds$$

$$\le \underbrace{(t - \xi) \cdot \|f(x(\xi))\|}_{=\alpha(t)} + \int_\xi^t \underbrace{L}_{=\beta(s)} \cdot \underbrace{\|x(s) - x(\xi)\|}_{=u(s)} \, ds$$

$$\le (t - \xi) \cdot \|f(x(\xi))\| \cdot e^{L(t-\xi)} \quad \text{via Grönwall}$$

And if $\xi \ge t$, we analogously get $\|x(t) - x(\xi)\| \le (\xi - t) \cdot \|f(x(\xi))\| \cdot e^{L(\xi-t)}$.

In particular, note that by the same method:

$$\|f(x(\xi))\| = \frac{1}{T}\|x(T) - x(0)\| \le \|f(x(0))\| e^{LT}$$

5. We use this result to bound the integral:

$$\frac{1}{T} \int_0^T \|\dot{x}(t) - c\|^2 \, dt \leq \frac{L^2}{T} \int_0^T \|x(t) - x(\xi)\|^2 \, dt$$

$$= \frac{L^2}{T} \left( \int_0^\xi \|x(t) - x(\xi)\|^2 \, dt + \int_\xi^T \|x(t) - x(\xi)\|^2 \, dt \right)$$

$$\leq \frac{L^2}{T} \|f(x(\xi))\|^2 \cdot \left( \int_0^\xi (\xi - t)^2 \cdot e^{2L(\xi - t)} \, dt + \int_\xi^T (t - \xi)^2 \cdot e^{2L(t - \xi)} \, dt \right)$$

$$\leq \ldots$$

$$\leq \|f(x(\xi))\|^2 \left( (2LT^2 - T + \frac{1}{2L}) e^{2LT} - \frac{1}{2L} \right)$$

$$\lesssim \|f(x(0))\|^2 e^{2LT} \mathcal{O}(LT^2 e^{2LT})$$

$$= \mathcal{O} \left( \|f(x(0))\|^2 LT e^{4LT} \right)$$

6. Plugging this back into the definition of the MGD, we get:

$$\text{MGD} = \sqrt{\frac{1}{T} \int_0^T \|\dot{x}(t) - c\|^2 \, dt} = \mathcal{O}(\sqrt{LT} e^{2LT})$$

# D  DATASET DESCRIPTION

Table 4: Optimized Datasets created with generated Python code published on Physiome. A t-unit of "—" refers to Physiome giving the time unit as dimensionless. We entered — in $\sigma_{dur}$, $\sigma_{state}$, $\sigma_{const}$ and JGD, if we were unable to create a dataset with our method due to errors thrown by the ode solver or the occurence of ode-explosions (see Section 4)

| Model | Domain | Channel | Constants | Time Unit | T | d-state | d-const | ATVL |
|---|---|---|---|---|---|---|---|---|
| DUP01 | Calcium-Dynamics | 2 | 13 | s | 30.0 | 0.3 | 0.05 | 2.697 |
| JEL01 | Endocrine | 2 | 15 | - | 30.0 | 0.5 | 0.05 | 2.58 |
| DOK01 | Electrophysiology | 18 | 46 | s | 10.0 | 0.1 | 0.05 | 2.277 |
| INA01 | Electrophysiology | 29 | 58 | s | 10.0 | 0.5 | 0.05 | 2.218 |
| WOL01 | Signal-Transduction | 9 | 19 | min | 10.0 | 0.3 | 0.05 | 1.973 |
| BOR01 | Calcium-Dynamics | 3 | 14 | min | 30.0 | 0.1 | 0.05 | 1.795 |
| HYN01 | Metabolism | 22 | 60 | min | 30.0 | 0.1 | 0.05 | 1.548 |
| JEL02 | Endocrine | 2 | 15 | - | 30.0 | 0.5 | 0.05 | 1.271 |
| DUP02 | Calcium-Dynamics | 3 | 19 | min | 30.0 | 0.1 | 0.05 | 1.202 |
| WOL02 | Metabolism | 7 | 14 | min | 30.0 | 0.1 | 0.1 | 0.895 |
| DIF01 | Electrophysiology | 16 | 50 | s | 10.0 | 0.1 | 0.1 | 0.735 |
| VAN01 | Metabolism | 10 | 38 | s | 10.0 | 0.1 | 0.05 | 0.407 |
| DUP03 | Calcium-Dynamics | 2 | 14 | s | 10.0 | 0.3 | 0.05 | 0.254 |
| BER01 | Signal-Transduction | 11 | 22 | ms | 30.0 | 0.1 | 0.3 | 0.179 |
| LEN01 | Endocrine | 3 | 7 | s | 30.0 | 0.1 | 0.1 | 0.178 |
| LI01 | Electrophysiology | 5 | 20 | s | 30.0 | 0.3 | 0.05 | 0.113 |
| LI02 | Electrophysiology | 5 | 20 | s | 10.0 | 0.1 | 0.05 | 0.097 |
| REV01 | Immunology | 5 | 10 | day | 30.0 | 0.5 | 0.3 | 0.081 |
| PUR01 | Neurobiology | 3 | 21 | ms | 30.0 | 0.1 | 0.05 | 0.049 |
| NYG01 | Electrophysiology | 29 | 51 | s | 0.33 | 0.5 | 0.1 | 0.047 |
| PUR02 | Neurobiology | 3 | 21 | ms | 30.0 | 0.1 | 0.05 | 0.044 |
| HOD01 | Electrophysiology | 4 | 8 | ms | 30.0 | 0.5 | 0.3 | 0.042 |
| REE01 | Metabolism | 4 | 21 | h | 30.0 | 0.5 | 0.05 | 0.035 |
| VIL01 | Signal-Transduction | 9 | 16 | h | 30.0 | 0.1 | 0.3 | 0.028 |
| KAR01 | Signal-Transduction | 16 | 21 | s | 10.0 | 0.3 | 0.05 | 0.023 |
| SHO01 | Excitation-contraction-Coupling | 56 | 105 | ms | 30.0 | 0.3 | 0.05 | 0.023 |
| BUT01 | Electrophysiology | 4 | 24 | ms | 30.0 | 0.3 | 0.05 | 0.02 |
| MAL01 | Endocrine | 9 | 51 | day | 30.0 | 0.5 | 0.05 | 0.019 |
| ASL01 | Electrophysiology | 30 | 83 | ms | 30.0 | 0.3 | 0.3 | 0.019 |
| BUT02 | Electrophysiology | 4 | 25 | ms | 30.0 | 0.3 | 0.05 | 0.016 |
| MIT01 | Immunology | 7 | 13 | day | 30.0 | 0.1 | 0.3 | 0.015 |
| GUP01 | Endocrine | 4 | 7 | h | 30.0 | 0.5 | 0.05 | 0.014 |
| GUY01 | Cardiovascular-Circulation | 2 | 9 | min | 30.0 | 0.3 | 0.05 | 0.013 |
| PHI01 | Circadian-Rhythms | 3 | 17 | h | 30.0 | 0.1 | 0.1 | 0.013 |
| GUY02 | Cardiovascular-Circulation | 2 | 11 | min | 30.0 | 0.3 | 0.05 | 0.013 |
| PUL01 | Cardiovascular-Circulation | 2 | 561 | min | 30.0 | 0.3 | 0.05 | 0.013 |
| CAL01 | Cell-Cycle | 17 | 43 | min | 30.0 | 0.5 | 0.1 | 0.013 |
| WOD01 | Immunology | 3 | 7 | s | 30.0 | 0.1 | 0.1 | 0.012 |
| GUP02 | Endocrine | 4 | 7 | h | 30.0 | 0.5 | 0.1 | 0.012 |
| M01 | Cardiovascular-Circulation | 2 | 558 | min | 30.0 | 0.3 | 0.05 | 0.012 |
| LEN02 | Endocrine | 3 | 12 | s | 30.0 | 0.5 | 0.1 | 0.012 |
| KAR02 | Signal-Transduction | 13 | 15 | s | 3.3 | 0.1 | 0.1 | 0.011 |
| SHO02 | Excitation-contraction-Coupling | 56 | 105 | ms | 10.0 | 0.3 | 0.05 | 0.011 |
| MAC01 | Ion-Transport | 7 | 35 | s | 0.33 | 0.5 | 0.05 | 0.01 |
| IRI01 | Electrophysiology | 23 | 69 | s | 0.33 | 0.3 | 0.1 | 0.01 |
| BAG01 | Signal-Transduction | 9 | 30 | s | 3.3 | 0.1 | 0.05 | 0.01 |
| WOL03 | Metabolism | 13 | 28 | min | 30.0 | 0.5 | 0.1 | 0.008 |
| WAN01 | Calcium-Dynamics | 5 | 27 | s | 30.0 | 0.3 | 0.1 | 0.008 |
| NEL01 | Immunology | 4 | 7 | day | 30.0 | 0.1 | 0.3 | 0.007 |
| HUA01 | Signal-Transduction | 18 | 37 | s | 30.0 | 0.1 | 0.3 | 0.007 |
| SVE01 | Cell-Cycle | 16 | 58 | min | 10.0 | 0.5 | 0.3 | 0.007 |
| GUY03 | Cardiovascular-Circulation | 2 | 8 | min | 30.0 | 0.5 | 0.05 | 0.006 |
| CIL01 | Cell-Cycle | 19 | 69 | min | 10.0 | 0.3 | 0.05 | 0.006 |
| LEL01 | Signal-Transduction | 10 | 38 | h | 30.0 | 0.5 | 0.05 | 0.006 |
| GAR01 | Cell-Cycle | 5 | 20 | min | 30.0 | 0.5 | 0.3 | 0.006 |
| COU01 | Electrophysiology | 21 | 49 | ms | 30.0 | 0.5 | 0.05 | 0.005 |
| AUT01 | Cardiovascular-Circulation | 2 | 570 | min | 3.3 | 0.5 | 0.05 | 0.005 |
| GUY04 | Cardiovascular-Circulation | 2 | 41 | min | 3.3 | 0.5 | 0.05 | 0.005 |
| RAZ01 | Myofilament-Mechanics | 3 | 19 | s | 30.0 | 0.1 | 0.1 | 0.005 |
| GRA01 | Pkpd | 9 | 30 | h | 10.0 | 0.5 | 0.05 | 0.005 |
| SNE01 | Calcium-Dynamics | 5 | 26 | s | 1.0 | 0.5 | 0.3 | 0.005 |
| WOD02 | Immunology | 10 | 14 | day | 10.0 | 0.3 | 0.1 | 0.005 |
| WOD03 | Immunology | 4 | 12 | day | 30.0 | 0.5 | 0.1 | 0.004 |
| NEU01 | Immunology | 3 | 8 | day | 10.0 | 0.3 | 0.3 | 0.004 |
| FAR01 | Signal-Transduction | 5 | 35 | h | 1.0 | 0.1 | 0.1 | 0.004 |
| SNE02 | Calcium-Dynamics | 3 | 13 | s | 30.0 | 0.3 | 0.1 | 0.004 |
| FAR02 | Signal-Transduction | 5 | 35 | h | 1.0 | 0.1 | 0.1 | 0.004 |

| Model | Domain | Channel | Constants | Time Unit | T | d-state | d-const | ATVL |
|---|---|---|---|---|---|---|---|---|
| BER02 | Electrophysiology | 7 | 47 | ms | 30.0 | 0.5 | 0.1 | 0.004 |
| GUY05 | Cardiovascular-Circulation | 3 | 13 | min | 30.0 | 0.3 | 0.05 | 0.004 |
| MCA01 | Electrophysiology | 10 | 13 | ms | 30.0 | 0.3 | 0.3 | 0.004 |
| MCA02 | Electrophysiology | 10 | 13 | ms | 30.0 | 0.3 | 0.3 | 0.004 |
| NOV01 | Cell-Cycle | 11 | 45 | min | 30.0 | 0.5 | 0.3 | 0.004 |
| YAM01 | Excitation-contraction-Coupling | 5 | 20 | s | 3.3 | 0.1 | 0.1 | 0.004 |
| RAL01 | Metabolism | 24 | 142 | min | 30.0 | 0.1 | 0.3 | 0.004 |
| VIS01 | Electrophysiology | 25 | 75 | ms | 10.0 | 0.5 | 0.05 | 0.004 |
| CAM01 | Myofilament-Mechanics | 4 | 10 | s | 30.0 | 0.3 | 0.05 | 0.004 |
| VIS02 | Electrophysiology | 25 | 75 | ms | 10.0 | 0.5 | 0.05 | 0.004 |
| SNY01 | Calcium-Dynamics | 10 | 27 | s | 30.0 | 0.5 | 0.3 | 0.003 |
| NOB01 | Electrophysiology | 20 | 69 | s | 0.33 | 0.1 | 0.05 | 0.003 |
| LEL02 | Signal-Transduction | 3 | 10 | h | 30.0 | 0.5 | 0.3 | 0.003 |
| MAL02 | Electrophysiology | 29 | 86 | ms | 30.0 | 0.5 | 0.05 | 0.003 |
| UED01 | Signal-Transduction | 10 | 55 | h | 30.0 | 0.5 | 0.1 | 0.002 |
| CHE01 | Cell-Cycle | 36 | 136 | min | 10.0 | 0.5 | 0.3 | 0.002 |
| POT01 | Endocrine | 33 | 97 | h | 30.0 | 0.1 | 0.1 | 0.002 |
| CHE02 | Cell-Cycle | 13 | 71 | min | 30.0 | 0.1 | 0.05 | 0.002 |
| PAR01 | Ion-Transport | 7 | 35 | s | 3.3 | 0.5 | 0.05 | 0.002 |
| JEL03 | Endocrine | 2 | 15 | - | 30.0 | 0.1 | 0.1 | 0.002 |
| GOO01 | Circadian-Rhythms | 2 | 6 | s | 30.0 | 0.5 | 0.3 | 0.002 |
| DIX01 | Immunology | 5 | 18 | day | 3.3 | 0.5 | 0.05 | 0.002 |
| CSI01 | Cell-Cycle | 14 | 93 | min | 10.0 | 0.5 | 0.1 | 0.002 |
| HAT01 | Signal-Transduction | 33 | 80 | s | 30.0 | 0.5 | 0.1 | 0.002 |
| VAS01 | Metabolism | 10 | 19 | s | 30.0 | 0.1 | 0.3 | 0.002 |
| MIT02 | Electrophysiology | 2 | 10 | ms | 30.0 | 0.3 | 0.3 | 0.002 |
| MIJ01 | Myofilament-Mechanics | 4 | 12 | s | 30.0 | 0.5 | 0.3 | 0.002 |
| REI01 | Signal-Transduction | 4 | 11 | s | 1.0 | 0.1 | 0.3 | 0.002 |
| GUY06 | Cardiovascular-Circulation | 5 | 45 | min | 3.3 | 0.3 | 0.05 | 0.002 |
| GAL01 | Electrophysiology | 3 | 28 | ms | 30.0 | 0.5 | 0.05 | 0.002 |
| LEM01 | Endocrine | 3 | 28 | day | 30.0 | 0.5 | 0.05 | 0.002 |
| SRI01 | Cell-Cycle | 6 | 25 | min | 30.0 | 0.5 | 0.05 | 0.001 |
| FAB01 | Electrophysiology | 25 | 81 | ms | 10.0 | 0.3 | 0.05 | 0.001 |
| CUI01 | Calcium-Dynamics | 4 | 22 | min | 3.3 | 0.1 | 0.3 | 0.001 |
| LEM02 | Endocrine | 3 | 28 | day | 30.0 | 0.5 | 0.05 | 0.001 |
| SCH01 | Immunology | 5 | 15 | day | 30.0 | 0.5 | 0.3 | 0.001 |
| HAU01 | Endocrine | 4 | 14 | min | 30.0 | 0.5 | 0.1 | 0.001 |
| BOR02 | Calcium-Dynamics | 3 | 15 | min | 0.33 | 0.3 | 0.05 | 0.001 |
| WOD04 | Immunology | 2 | 5 | s | 30.0 | 0.5 | 0.3 | 0.001 |
| FIN01 | Electrophysiology | 27 | 61 | ms | 0.33 | 0.1 | 0.3 | 0.001 |
| CHA01 | Metabolism | 12 | 43 | min | 0.33 | 0.1 | 0.3 | 0.001 |
| HAL01 | Cell-Cycle | 4 | 7 | month | 30.0 | 0.1 | 0.05 | 0.001 |
| BAR01 | Cell-Cycle | 34 | 55 | min | 3.3 | 0.1 | 0.05 | 0.001 |
| RAP01 | Calcium-Dynamics | 11 | 57 | h | 10.0 | 0.5 | 0.05 | 0.001 |
| KOM01 | Immunology | 2 | 9 | h | 30.0 | 0.5 | 0.1 | 0.001 |
| NOV02 | Cell-Cycle | 9 | 31 | min | 10.0 | 0.5 | 0.05 | 0.001 |
| NOV03 | Cell-Cycle | 13 | 43 | min | 10.0 | 0.5 | 0.05 | 0.001 |
| VEM01 | Signal-Transduction | 17 | 33 | s | 30.0 | 0.5 | 0.3 | 0.001 |
| MIF01 | Electrophysiology | 5 | 14 | ms | 0.33 | 0.1 | 0.3 | 0.001 |
| COR01 | Electrophysiology | 14 | 51 | - | 30.0 | 0.5 | 0.05 | 0.001 |
| BON01 | Immunology | 3 | 7 | day | 30.0 | 0.5 | 0.3 | 0.001 |
| YAT01 | Cell-Cycle | 2 | 5 | day | 30.0 | 0.5 | 0.05 | 0.001 |
| BON02 | Immunology | 4 | 13 | day | 30.0 | 0.5 | 0.3 | 0.001 |
| BON03 | Immunology | 3 | 8 | day | 30.0 | 0.5 | 0.3 | 0.001 |
| BRO01 | Endocrine | 2 | 11 | min | 30.0 | 0.5 | 0.3 | 0.0 |
| IZA01 | Excitation-contraction-Coupling | 2 | 30 | ms | 10.0 | 0.1 | 0.3 | 0.0 |
| VIL02 | Signal-Transduction | 6 | 9 | min | 30.0 | 0.5 | 0.05 | 0.0 |
| BER03 | Electrophysiology | 4 | 21 | ms | 30.0 | 0.1 | 0.3 | 0.0 |
| BER04 | Electrophysiology | 4 | 21 | ms | 30.0 | 0.1 | 0.3 | 0.0 |
| BER05 | Electrophysiology | 5 | 32 | ms | 30.0 | 0.5 | 0.05 | 0.0 |
| KOM02 | Endocrine | 3 | 13 | day | 30.0 | 0.1 | 0.1 | 0.0 |
| LEN03 | Endocrine | 5 | 15 | s | 30.0 | 0.5 | 0.3 | 0.0 |
| RAT01 | Endocrine | 3 | 13 | - | 30.0 | 0.5 | 0.1 | 0.0 |
| BUE01 | Electrophysiology | 4 | 32 | ms | 30.0 | 0.5 | 0.1 | 0.0 |
| GUY07 | Cardiovascular-Circulation | 4 | 27 | min | 30.0 | 0.5 | 0.05 | 0.0 |
| HUN01 | Electrophysiology | 29 | 70 | ms | 1.0 | 0.1 | 0.1 | 0.0 |
| ROM01 | Cell-Cycle | 6 | 28 | min | 30.0 | 0.5 | 0.1 | 0.0 |
| SHO03 | Calcium-Dynamics | 7 | 40 | ms | 1.0 | 0.1 | 0.1 | 0.0 |
| GUY08 | Cardiovascular-Circulation | 2 | 17 | min | 30.0 | 0.5 | 0.05 | 0.0 |
| POT02 | Endocrine | 5 | 12 | s | 30.0 | 0.5 | 0.05 | 0.0 |
| CIR01 | Cardiovascular-Circulation | 44 | 96 | ms | 30.0 | 0.1 | 0.05 | 0.0 |
| STE01 | Excitation-contraction-Coupling | 4 | 5 | ms | 30.0 | 0.5 | 0.3 | 0.0 |
| PED01 | Endocrine | 9 | 54 | ms | 10.0 | 0.3 | 0.05 | 0.0 |
| GUY09 | Cardiovascular-Circulation | 2 | 5 | min | 30.0 | 0.1 | 0.3 | 0.0 |
| GON01 | Endocrine | 3 | 11 | min | 0.33 | 0.1 | 0.05 | 0.0 |
| ELE01 | Cardiovascular-Circulation | 4 | 566 | min | 30.0 | 0.1 | 0.05 | 0.0 |
| GUY10 | Cardiovascular-Circulation | 4 | 17 | min | 30.0 | 0.1 | 0.3 | 0.0 |

Continued on next page

| Model | Domain | Channel | Constants | Time Unit | T | d-state | d-const | ATVL |
|---|---|---|---|---|---|---|---|---|
| POT03 | Endocrine | 5 | 16 | s | 30.0 | 0.1 | 0.3 | 0.0 |
| CLO01 | Metabolism | 7 | 42 | s | 0.33 | 0.1 | 0.1 | 0.0 |
| CLO02 | Metabolism | 7 | 42 | s | 0.33 | 0.1 | 0.1 | 0.0 |
| RAN01 | Signal-Transduction | 32 | 32 | s | 30.0 | 0.1 | 0.3 | 0.0 |
| HYP01 | Cardiovascular-Circulation | 3 | 560 | min | 30.0 | 0.1 | 0.3 | 0.0 |
| GUY11 | Cardiovascular-Circulation | 3 | 10 | min | 30.0 | 0.1 | 0.3 | 0.0 |
| SED01 | Signal-Transduction | 20 | 33 | min | - | - | - | - |
| LIV01 | Electrophysiology | 18 | 68 | ms | - | - | - | - |
| KAT01 | Electrophysiology | 9 | 54 | s | - | - | - | - |
| CUI02 | Electrophysiology | 20 | 17 | s | - | - | - | - |
| NIE01 | Electrophysiology | 26 | 188 | ms | - | - | - | - |
| CHA02 | Electrophysiology | 12 | 38 | s | - | - | - | - |
| KAT02 | Electrophysiology | 9 | 54 | ms | - | - | - | - |
| FAV01 | Electrophysiology | 80 | 770 | s | - | - | - | - |
| CHA03 | Electrophysiology | 9 | 27 | s | - | - | - | - |
| CLO03 | Neurobiology | 36 | 125 | s | - | - | - | - |
| VIN01 | Metabolism | 25 | 342 | min | - | - | - | - |
| DEM01 | Signal-Transduction | 29 | 55 | s | - | - | - | - |
| CHE03 | Signal-Transduction | 14 | 20 | s | - | - | - | - |
| COO01 | Signal-Transduction | 13 | 38 | s | - | - | - | - |
| NAZ01 | Metabolism | 8 | 55 | - | - | - | - | - |
| HUN02 | Electrophysiology | 29 | 69 | ms | - | - | - | - |
| NAZ02 | Metabolism | 8 | 25 | s | - | - | - | - |
| WAN02 | Signal-Transduction | 4 | 4 | s | - | - | - | - |
| CUR01 | Metabolism | 0 | 21 | | - | - | - | - |
| DAS01 | Metabolism | 0 | 30 | | - | - | - | - |
| VIN02 | Metabolism | 25 | 342 | min | - | - | - | - |
| HEY01 | Metabolism | 11 | 18 | s | - | - | - | - |
| CHE04 | Signal-Transduction | 14 | 20 | s | - | - | - | - |
| PRO01 | Signal-Transduction | 14 | 24 | s | - | - | - | - |
| TRA01 | Metabolism | 11 | 71 | ms | - | - | - | - |
| BEA01 | Metabolism | 19 | 60 | s | - | - | - | - |
| BEN01 | Electrophysiology | 29 | 69 | ms | - | - | - | - |
| NOB02 | Electrophysiology | 22 | 70 | s | - | - | - | - |
| LI03 | Electrophysiology | 12 | 37 | s | - | - | - | - |
| THA01 | Pkpd | 2 | 15 | week | - | - | - | - |
| MAC02 | Endocrine | 4 | 14 | min | - | - | - | - |
| MAI01 | Cardiovascular-Circulation | 7 | 58 | s | - | - | - | - |
| GUY12 | Cardiovascular-Circulation | 43 | 210 | min | - | - | - | - |
| MAI02 | Cardiovascular-Circulation | 3 | 21 | s | - | - | - | - |
| KYR01 | Endocrine | 5 | 24 | min | - | - | - | - |
| MAC03 | Endocrine | 7 | 28 | min | - | - | - | - |
| BAY01 | Calcium-Dynamics | 3 | 2 | s | - | - | - | - |
| RIC01 | Signal-Transduction | 8 | 11 | s | - | - | - | - |
| BEN02 | Mechanical-Constitutive-Laws | 7 | 26 | s | - | - | - | - |
| PAN01 | Excitation-contraction-Coupling | 22 | 93 | ms | - | - | - | - |
| MAI03 | Cardiovascular-Circulation | 7 | 32 | s | - | - | - | - |
| OVE01 | Pkpd | 3 | 37 | h | - | - | - | - |
| PMR01 | Gene-Regulation | 0 | 0 | | - | - | - | - |
| GUY13 | Cardiovascular-Circulation | 0 | 10 | | - | - | - | - |
| MOD01 | Cardiovascular-Circulation | 14 | 88 | s | - | - | - | - |
| PMR02 | Gene-Regulation | 0 | 0 | | - | - | - | - |
| GUY14 | Cardiovascular-Circulation | 4 | 72 | min | - | - | - | - |
| ESP01 | Electrophysiology | 21 | 70 | s | - | - | - | - |
| BEN03 | Electrophysiology | 29 | 69 | ms | - | - | - | - |
| BOY01 | Electrophysiology | 22 | 90 | s | - | - | - | - |
| ESP02 | Electrophysiology | 21 | 70 | s | - | - | - | - |
| ASL02 | Electrophysiology | 29 | 49 | s | - | - | - | - |
| OST01 | Electrophysiology | 8 | 31 | s | - | - | - | - |
| MIC01 | Electrophysiology | 43 | 97 | s | - | - | - | - |
| NIE02 | Electrophysiology | 5 | 25 | ms | - | - | - | - |
| INA02 | Electrophysiology | 29 | 63 | s | - | - | - | - |
| MAH01 | Electrophysiology | 26 | 80 | ms | - | - | - | - |
| SED02 | Signal-Transduction | 21 | 38 | min | - | - | - | - |

# E  LINKS OF ODE MODELS

Table 5: Model abbreviations with links to their Physiome page. The Physiome page contains information about the ODE model, the respective publication. Furthermore, the generated Python code can also be found there.

| Abbreviation | Link |
| --- | --- |
| ASL01 | https://models.physiomeproject.org/exposure/bb75b3021730966e2827967a66188cb2 |
| ASL02 | https://models.physiomeproject.org/exposure/0e3a603db8f464ae89becb8d89225d90 |
| AUT01 | https://models.physiomeproject.org/exposure/827af05888f8e152f448d9cd8c6a8d09 |
| BAG01 | https://models.physiomeproject.org/exposure/98c3ef8d8c8f444d3a492fe777b4f6d0 |
| BAR01 | https://models.physiomeproject.org/exposure/455155cfef38300290a75bcfe7440c31 |
| BAY01 | https://models.physiomeproject.org/exposure/210f6601f6461be8443592ff071d2592 |
| BEA01 | https://models.physiomeproject.org/exposure/959a4da5d8a0638f4b36adf5800f4fc0 |
| BEN01 | https://models.physiomeproject.org/exposure/5d2fd39aa16f562d3bc19119847b7db0 |
| BEN02 | https://models.physiomeproject.org/exposure/b503501533abcf0e70786789f08cb902 |
| BEN03 | https://models.physiomeproject.org/exposure/5d2fd39aa16f562d3bc19119847b7db0 |
| BER01 | https://models.physiomeproject.org/exposure/c6d3b19b455fbb3f80813337ac127674 |
| BER02 | https://models.physiomeproject.org/exposure/a7f8b4574fe4802f1e06f247006962ac |
| BER03 | https://models.physiomeproject.org/exposure/fe91f06c8311ffd90fd1eaabe9b0032a |
| BER04 | https://models.physiomeproject.org/exposure/fe91f06c8311ffd90fd1eaabe9b0032a |
| BER05 | https://models.physiomeproject.org/exposure/48e767ee37f347e1a3876d0549a0866b |
| BON01 | https://models.physiomeproject.org/exposure/def2cd06391e17d8966e22997a65476a |
| BON02 | https://models.physiomeproject.org/exposure/def2cd06391e17d8966e22997a65476a |
| BON03 | https://models.physiomeproject.org/exposure/def2cd06391e17d8966e22997a65476a |
| BOR01 | https://models.physiomeproject.org/exposure/7ba127c00a5524c49f0a57e6589eda03 |
| BOR02 | https://models.physiomeproject.org/exposure/7ba127c00a5524c49f0a57e6589eda03 |
| BOY01 | https://models.physiomeproject.org/exposure/d4d2febdd2a9556eca88ceeac66bf696 |
| BRO01 | https://models.physiomeproject.org/exposure/1f40088c3eb67895ff8dcc5010d62bf1 |
| BUE01 | https://models.physiomeproject.org/exposure/954421556db2a0c7de89ce0bdc26bed9 |
| BUT01 | https://models.physiomeproject.org/exposure/293a909eeeca07d0ca8ad583839eb0bc |
| BUT02 | https://models.physiomeproject.org/exposure/293a909eeeca07d0ca8ad583839eb0bc |
| CAL01 | https://models.physiomeproject.org/exposure/1a3f36d015121d5596565fe7d9afb332 |
| CAM01 | https://models.physiomeproject.org/exposure/62183706711e435ff002b46088540850 |
| CHA01 | https://models.physiomeproject.org/exposure/a74fd808df7ba454945e5ac42486e3ad |
| CHA02 | https://models.physiomeproject.org/exposure/1f4c6ffdb5120ed09675e4c9d346952f |
| CHA03 | https://models.physiomeproject.org/exposure/7fce899159b8698ccb2c8569d19a0ea1 |
| CHE01 | https://models.physiomeproject.org/exposure/0e2337ffa4c550dd077c629b9e76330e |
| CHE02 | https://models.physiomeproject.org/exposure/635c1359f216ab8d064dfe8e8d3387b8 |
| CHE03 | https://models.physiomeproject.org/exposure/01f8729559c3985a3d09ff4d983da75a |
| CHE04 | https://models.physiomeproject.org/exposure/01f8729559c3985a3d09ff4d983da75a |
| CIL01 | https://models.physiomeproject.org/exposure/f5be28698749563c5564d1a4fa5c0446 |
| CIR01 | https://models.physiomeproject.org/exposure/ff8be5f140e68612284488cf9879eb5f |
| CLO01 | https://models.physiomeproject.org/exposure/e5cfb42225d4534a1e08979e57cf8bdd |
| CLO02 | https://models.physiomeproject.org/exposure/e5cfb42225d4534a1e08979e57cf8bdd |
| CLO03 | https://models.physiomeproject.org/exposure/6e249a04f5c751e42ba41504d84e6e49 |
| COO01 | https://models.physiomeproject.org/exposure/39fee05138f428413db61aa29ff6f0de |
| COR01 | https://models.physiomeproject.org/exposure/9477b42c85e81f7da21059baa509ae21 |
| COU01 | https://models.physiomeproject.org/exposure/0e03bbe01606be5811691f9d5de10b65 |
| CSI01 | https://models.physiomeproject.org/exposure/dada9c9d39d0783375d42f12f11dddf3 |
| CUI01 | https://models.physiomeproject.org/exposure/a3fb8998a3dd86ca914e55d60b3b7a52 |
| CUI02 | https://models.physiomeproject.org/exposure/45ddea4fcde109d7a2bef806aeae84d7 |
| CUR01 | https://models.physiomeproject.org/exposure/e08bd642df06eb59404c8e46afbb04a4 |
| DAS01 | https://models.physiomeproject.org/exposure/bb0ee0e6d141a529fa896ff3785f0eff |
| DEM01 | https://models.physiomeproject.org/exposure/32c9e9739454b40b5ba2d9cabb1fd079 |
| DIF01 | https://models.physiomeproject.org/exposure/91d93b61d7da56b6baf1f0c4d88ecd77 |
| DIX01 | https://models.physiomeproject.org/exposure/cb02cc9ce006eb44b2077111c1d548d2 |
| DOK01 | https://models.physiomeproject.org/exposure/462ab10275dfc099166c8a0e4f9e1be3 |
| DUP01 | https://models.physiomeproject.org/exposure/060c119cc5365e3d9cd0203c82fe0121 |
| DUP02 | https://models.physiomeproject.org/exposure/04cc3dc1c0c8e5dc3634f0220904d297 |
| DUP03 | https://models.physiomeproject.org/exposure/060c119cc5365e3d9cd0203c82fe0121 |
| ELE01 | https://models.physiomeproject.org/exposure/239e0bdec0c7054e0727fc111a54658c |
| ESP01 | https://models.physiomeproject.org/exposure/3ead51a9899108b4f8c8177c0b5f2421 |
| ESP02 | https://models.physiomeproject.org/exposure/3ead51a9899108b4f8c8177c0b5f2421 |
| FAB01 | https://models.physiomeproject.org/exposure/55643f2114a2a463ada007deb9fc3913 |
| FAR01 | https://models.physiomeproject.org/exposure/fe9ed942c523777274fcdb43c6be75f7 |
| FAR02 | https://models.physiomeproject.org/exposure/fe9ed942c523777274fcdb43c6be75f7 |
| FAV01 | https://models.physiomeproject.org/exposure/f17a3cade0f0f7986b060b8bbf5c2957 |
| FIN01 | https://models.physiomeproject.org/exposure/eeb81adc372c2f172399ec7160b0331e |
| GAL01 | https://models.physiomeproject.org/exposure/bc0d80f7e1ccb2c288119f6e8fdfbc20 |
| GAR01 | https://models.physiomeproject.org/exposure/4f5fb67ce413238e24e5277f46fd5d21 |
| GON01 | https://models.physiomeproject.org/exposure/07464e63da82057cc6e44d33ec60cd7c |
| GOO01 | https://models.physiomeproject.org/exposure/b69b4d821f5b2996cda5f2b4a3e9ca8f |
| GRA01 | https://models.physiomeproject.org/exposure/32d3dcc3ab074d17905c10f2ba26b54f |
| GUP01 | https://models.physiomeproject.org/exposure/9b91dc35df3c4ca3ee14f6a3f4191db7 |

| Abbreviation | Link |
| --- | --- |
| GUP02 | https://models.physiomeproject.org/exposure/9b91dc35df3c4ca3ee14f6a3f4191db7 |
| GUY01 | https://models.physiomeproject.org/exposure/d20c1a1db1b4027605b87b4361e62560 |
| GUY02 | https://models.physiomeproject.org/exposure/63f124c018745aa8ec92a091434f3dfc |
| GUY03 | https://models.physiomeproject.org/exposure/c1461e469a5e282face2fdf37817ca10 |
| GUY04 | https://models.physiomeproject.org/exposure/827af05888f8e152f448d9cd8c6a8d09 |
| GUY05 | https://models.physiomeproject.org/exposure/be69ef6c1a42264162ae6e370e6107b5 |
| GUY06 | https://models.physiomeproject.org/exposure/e1f152d0dbd1dec4111faa117db85bc0 |
| GUY07 | https://models.physiomeproject.org/exposure/f3272a51c6e95c70eb1309d30c08d4cf |
| GUY08 | https://models.physiomeproject.org/exposure/f97a5eb092b12f4f0f32ac51ee20d20e |
| GUY09 | https://models.physiomeproject.org/exposure/64f53aa4552f15dc6805444ba0559c5a |
| GUY10 | https://models.physiomeproject.org/exposure/239e0bdec0c7054e0727fc111a54658c |
| GUY11 | https://models.physiomeproject.org/exposure/ea5144be46b69ba503a2d0cd5cbd5b96 |
| GUY12 | https://models.physiomeproject.org/exposure/cd10322c000e6ff64441464f8773ed83 |
| GUY13 | https://models.physiomeproject.org/exposure/4c73baa30cb81cf0cd83b4e8d42a5358 |
| GUY14 | https://models.physiomeproject.org/exposure/b897ad1b96031d40293b2e2e10684ffe |
| HAL01 | https://models.physiomeproject.org/exposure/77bdebf1280135c36f94cac077f74e18 |
| HAT01 | https://models.physiomeproject.org/exposure/c14e77a831ec3f1e56ef03240986a8b7 |
| HAU01 | https://models.physiomeproject.org/exposure/f9b96930ad2e7f5c9e15b1169a9378c7 |
| HEY01 | https://models.physiomeproject.org/exposure/3c875389313cd73f120d69a59cc65acb |
| HOD01 | https://models.physiomeproject.org/exposure/5d116522c3b43ccaeb87a1ed10139016 |
| HUA01 | https://models.physiomeproject.org/exposure/3227298fc9cc3afc14d058750142c69c |
| HUN01 | https://models.physiomeproject.org/exposure/f4b7120aa512c7f5e7a0664abcee3e8b |
| HUN02 | https://models.physiomeproject.org/exposure/f4b7120aa512c7f5e7a0664abcee3e8b |
| HYN01 | https://models.physiomeproject.org/exposure/b5c9ac7a7a8b76918c02b39967ae191b |
| HYP01 | https://models.physiomeproject.org/exposure/ea5144be46b69ba503a2d0cd5cbd5b96 |
| INA01 | https://models.physiomeproject.org/exposure/08bcead2dc05cf2709a598e7f61a6182 |
| INA02 | https://models.physiomeproject.org/exposure/08bcead2dc05cf2709a598e7f61a6182 |
| IRI01 | https://models.physiomeproject.org/exposure/7af88a567b3fdbb5209ddd2eee44075c |
| IZA01 | https://models.physiomeproject.org/exposure/94a5dddfa94d46f1d842e4e87a94646f |
| JEL01 | https://models.physiomeproject.org/exposure/4817bdfdd245412f52dd79da59137fe8 |
| JEL02 | https://models.physiomeproject.org/exposure/4817bdfdd245412f52dd79da59137fe8 |
| JEL03 | https://models.physiomeproject.org/exposure/4817bdfdd245412f52dd79da59137fe8 |
| KAR01 | https://models.physiomeproject.org/exposure/b5ddf62f58c911e794174d61f9aedf61 |
| KAR02 | https://models.physiomeproject.org/exposure/46a466d519479cee832ebf8b66c1ea00 |
| KAT01 | https://models.physiomeproject.org/exposure/2a3e1e865606c4da94c251017038f820 |
| KAT02 | https://models.physiomeproject.org/exposure/2a3e1e865606c4da94c251017038f820 |
| KOM01 | https://models.physiomeproject.org/exposure/88ec4bc9780f123780b75e4c9ce1682b |
| KOM02 | https://models.physiomeproject.org/exposure/fa7996b256f19f92568fb7ad414b6cba |
| KYR01 | https://models.physiomeproject.org/exposure/fd3eeaae0cbb662a9d5a2b47ce81a825 |
| LEL01 | https://models.physiomeproject.org/exposure/1db3a968ce8585201acd753ce857002a |
| LEL02 | https://models.physiomeproject.org/exposure/1db3a968ce8585201acd753ce857002a |
| LEM01 | https://models.physiomeproject.org/exposure/3094a5c3e0810028a9accc3b773cab36 |
| LEM02 | https://models.physiomeproject.org/exposure/3094a5c3e0810028a9accc3b773cab36 |
| LEN01 | https://models.physiomeproject.org/exposure/fb612e6ce429bc45d99388c045a100cb |
| LEN02 | https://models.physiomeproject.org/exposure/520cbff71e195d0a0ed36ce1c78d46d5 |
| LEN03 | https://models.physiomeproject.org/exposure/520cbff71e195d0a0ed36ce1c78d46d5 |
| LI01 | https://models.physiomeproject.org/exposure/dfe4f6c90d58266f0f5d6d320c291e40 |
| LI02 | https://models.physiomeproject.org/exposure/dfe4f6c90d58266f0f5d6d320c291e40 |
| LI03 | https://models.physiomeproject.org/exposure/dfe4f6c90d58266f0f5d6d320c291e40 |
| LIV01 | https://models.physiomeproject.org/exposure/dedfcd1a135ddb59e9d979ec1376a44f |
| M01 | https://models.physiomeproject.org/exposure/d20c1a1db1b4027605b87b4361e62560 |
| MAC01 | https://models.physiomeproject.org/exposure/a499a7082706164315ad07feff408850 |
| MAC02 | https://models.physiomeproject.org/exposure/1229d793e4a010088736d682bd35d463 |
| MAC03 | https://models.physiomeproject.org/exposure/1229d793e4a010088736d682bd35d463 |
| MAH01 | https://models.physiomeproject.org/exposure/a5586b72d07ce03fc40fc98ee846d7a5 |
| MAI01 | https://models.physiomeproject.org/exposure/a195d957a2d63ac4defe3232fd0ea50c |
| MAI02 | https://models.physiomeproject.org/exposure/9970597960dbe3a381d773d90c0298c2 |
| MAI03 | https://models.physiomeproject.org/exposure/feccb0f0649425a61526049f35c3b7fb |
| MAL01 | https://models.physiomeproject.org/exposure/8e081783cbe18d346eaf14cd5c1ae18f |
| MAL02 | https://models.physiomeproject.org/exposure/3c1c7b17df06921a4e1b05c639a45d32 |
| MCA01 | https://models.physiomeproject.org/exposure/60e23c3228a3e455699846704006a8fe |
| MCA02 | https://models.physiomeproject.org/exposure/60e23c3228a3e455699846704006a8fe |
| MIC01 | https://models.physiomeproject.org/exposure/9e77fb28778473706ebd43bd5bea86d7 |
| MIF01 | https://models.physiomeproject.org/exposure/8854e495f7e4240b1607820088b13138 |
| MIJ01 | https://models.physiomeproject.org/exposure/a9cd3fd1ec3c3bbe2613ca9e7490a367 |
| MIT01 | https://models.physiomeproject.org/exposure/0d32bf39c8af51c44373b2e68c3cec74 |
| MIT02 | https://models.physiomeproject.org/exposure/9efc295a6f3360044b4712c28b8314e7 |
| MOD01 | https://models.physiomeproject.org/exposure/c49d416ae3a5132882e6ea7479ba50f5 |
| NAZ01 | https://models.physiomeproject.org/exposure/45de427094d4814e74156890fd99fcc6 |
| NAZ02 | https://models.physiomeproject.org/exposure/45de427094d4814e74156890fd99fcc6 |
| NEL01 | https://models.physiomeproject.org/exposure/e811b8311a550fb0eedf95402b3166d0 |
| NEU01 | https://models.physiomeproject.org/exposure/a7e3d6ea01b67ecbe06e11f7909bcb93 |
| NIE01 | https://models.physiomeproject.org/exposure/59a44249dec83576d97fd3fce46ec5f9 |
| NIE02 | https://models.physiomeproject.org/exposure/97fb1de5199b1a74c89281db97aecc13 |
| NOB01 | https://models.physiomeproject.org/exposure/e42fb39e149f444be872e04e6e731ac0 |
| NOB02 | https://models.physiomeproject.org/exposure/a40c4434423c0436e2789a2d457b7ab2 |
| NOV01 | https://models.physiomeproject.org/exposure/ea5f37b01b27c7ba25b6c179a12a464a |

| Abbreviation | Link |
|---|---|
| NOV02 | https://models.physiomeproject.org/exposure/1e1bee6ef3243503e7e1531cfd61bb3f |
| NOV03 | https://models.physiomeproject.org/exposure/e24887f982e9246d05ba0f7152bd4aaa |
| NYG01 | https://models.physiomeproject.org/exposure/ad761ce160f3b4077bbae7a004c229e3 |
| OST01 | https://models.physiomeproject.org/exposure/d9de93b128da322a4d50f24589980ea1 |
| OVE01 | https://models.physiomeproject.org/exposure/504e7681708fc7b1260db25363658be5 |
| PAR01 | https://models.physiomeproject.org/exposure/a07c660076d3caddac2d136b0c0c7c05 |
| PED01 | https://models.physiomeproject.org/exposure/00bc5ddf4f576af8f81ffc429a183a73 |
| PHI01 | https://models.physiomeproject.org/exposure/87ef0e70902ed8a49d10252617835aa9 |
| PMR01 | https://models.physiomeproject.org/exposure/f2a5a551e8e22f18e5adb641243269c5 |
| PMR01 | https://models.physiomeproject.org/exposure/62b7b4336d5d1caa0a01c3faff8b33ea |
| PMR01 | https://models.physiomeproject.org/exposure/2fa5de12b0923874aee04486617eb2bd |
| PMR02 | https://models.physiomeproject.org/exposure/52e2a146e8bf5976379803fec46138a7 |
| PMR02 | https://models.physiomeproject.org/exposure/29a0ec2468a49a64a123f927083260f0 |
| POT01 | https://models.physiomeproject.org/exposure/05510be013e1d096e26c3716c950712d |
| POT02 | https://models.physiomeproject.org/exposure/80982c99e643a576d10e7f3e271a2299 |
| POT03 | https://models.physiomeproject.org/exposure/80982c99e643a576d10e7f3e271a2299 |
| PRO01 | https://models.physiomeproject.org/exposure/026ccefa76a5ea437d6339aae90c2d37 |
| PUL01 | https://models.physiomeproject.org/exposure/63f124c018745aa8ec92a091434f3dfc |
| PUR01 | https://models.physiomeproject.org/exposure/baa78a664bd1f620acce7257883fc91a |
| PUR02 | https://models.physiomeproject.org/exposure/baa78a664bd1f620acce7257883fc91a |
| RAL01 | https://models.physiomeproject.org/exposure/d8dae90ac67ab8b892864c7c66bf6110 |
| RAN01 | https://models.physiomeproject.org/exposure/9fb32c29b3650b853590d30c7e3374af |
| RAP01 | https://models.physiomeproject.org/exposure/ea84bd6e27e521677bfc2d6283b371f3 |
| RAT01 | https://models.physiomeproject.org/exposure/f9611ad158302284299c1020faf878ee |
| RAZ01 | https://models.physiomeproject.org/exposure/6703dcc072ed1c21cade028b9abbcfa0 |
| REE01 | https://models.physiomeproject.org/exposure/b0a3d0dbb029b6cbd2a193014b779a82 |
| REI01 | https://models.physiomeproject.org/exposure/cfbc61b251bcccb085ffd9d074c1d44c |
| REV01 | https://models.physiomeproject.org/exposure/03852e75c08161cf63d70417ec4a2fae |
| RIC01 | https://models.physiomeproject.org/exposure/e340f005288ec6bd49535cf792769dd0 |
| ROM01 | https://models.physiomeproject.org/exposure/9a75ea45513cec9ac59991afa40430a4 |
| SCH01 | https://models.physiomeproject.org/exposure/926ad168d96e3fe4b87f452890631f86 |
| SED01 | https://models.physiomeproject.org/exposure/c18a94e82ef7ed7c9dad399e61677b5e |
| SED02 | https://models.physiomeproject.org/exposure/c18a94e82ef7ed7c9dad399e61677b5e |
| SHO03 | https://models.physiomeproject.org/exposure/e10def01073e56fb32f06bf59e66a318 |
| SNE01 | https://models.physiomeproject.org/exposure/4f67bf3d82a1ad07d332e383937041e2 |
| SNE02 | https://models.physiomeproject.org/exposure/88053fd5c8867b72ee23c1702097e377 |
| SNY01 | https://models.physiomeproject.org/exposure/8d791fd6aba9793fb0149d5b6e3afcee |
| SRI01 | https://models.physiomeproject.org/exposure/500b72e21302febd0f3dc746dc22af81 |
| STE01 | https://models.physiomeproject.org/exposure/b060fdbcfae8c7d85e595c24d36ab11b |
| SVE01 | https://models.physiomeproject.org/exposure/aad2ca9cce58dba12c8d9d85dccada0f |
| THA01 | https://models.physiomeproject.org/exposure/1985e7c820ff102b1caebc241df65ce7 |
| TRA01 | https://models.physiomeproject.org/exposure/cfa7684fb084ad748bf3061569d99334 |
| UED01 | https://models.physiomeproject.org/exposure/738882ee938d6b2e2d264714259e0416 |
| VAN01 | https://models.physiomeproject.org/exposure/a0df3e434c6d37073783883cb4dff2d3 |
| VAS01 | https://models.physiomeproject.org/exposure/05557c5c5ae347b0a6099f66eb33e50f |
| VEM01 | https://models.physiomeproject.org/exposure/c1e5f2110188582f6f44f85930733144 |
| VIL01 | https://models.physiomeproject.org/exposure/d5af7e23b0b67c042ca76949c7e59c79 |
| VIL02 | https://models.physiomeproject.org/exposure/882bd564f637b7a506fafdf8a953a506 |
| VIN01 | https://models.physiomeproject.org/exposure/07cdb663179c77de23e425a56464a2aa |
| VIN02 | https://models.physiomeproject.org/exposure/07cdb663179c77de23e425a56464a2aa |
| VIS01 | https://models.physiomeproject.org/exposure/4c26896c035fd04e70658d792affad9d |
| VIS02 | https://models.physiomeproject.org/exposure/4c26896c035fd04e70658d792affad9d |
| WAN01 | https://models.physiomeproject.org/exposure/a9e94f2a37908bf3ebb1d6aa4bef82c4 |
| WAN02 | https://models.physiomeproject.org/exposure/a93163783c79b766b9ab88c8a20c1e41 |
| WOD01 | https://models.physiomeproject.org/exposure/eb9280eef64afce83878426e24d5b31c |
| WOD02 | https://models.physiomeproject.org/exposure/9fddc0e3b1bb1d6ad86baeea2e053715 |
| WOD03 | https://models.physiomeproject.org/exposure/869cfba0248523dcd57a7e49185f70ab |
| WOD04 | https://models.physiomeproject.org/exposure/eb9280eef64afce83878426e24d5b31c |
| WOL01 | https://models.physiomeproject.org/exposure/c7801370ba2f7c3e19ad4d12d55d4968 |
| WOL02 | https://models.physiomeproject.org/exposure/c1821457cc8c8135f3317ec4882b719f |
| WOL03 | https://models.physiomeproject.org/exposure/f68c97d153bb0e4a050cb4e143b047a8 |
| YAM01 | https://models.physiomeproject.org/exposure/5ad3bc65fe5ed254806310e2dfb1c5cb |
| YAT01 | https://models.physiomeproject.org/exposure/0b98c29503e70a6b35be0ed0f5719458 |

## F HYPERPARAMETER SEARCH

We exclusively conduct hyperparameter search on fold 0. For **GraFITi** (Yalavarthi et al., 2024) the hyperparameters for the search are as follows:

- The number of layers, with possible values [1, 2, 3, 4].
- The number of attention heads, with possible values [1, 2, 4].
- The latent dimension, with possible values [16, 32, 64, 128, 256].

For the **LinODEnet** model (Scholz et al., 2022) we search the hyperparameters from:

- The hidden dimension, with possible values [16, 32, 64, 128].
- The latent dimension, with possible values [64, 128, 192, 256].

For **GRU-ODE-Bayes** (De Brouwer et al., 2019) we tune the hidden size from [16, 32, 64, 128, 256]

For **Neural Flows** (Biloš et al., 2021) we define the hyperparameter spaces for the search are as follows:

- The number of flow layers, with possible values [1, 2, 4].
- The hidden dimension, with possible values [16, 32, 64, 128, 256].
- The flow model type, with possible values [GRU, ResNet].

For the **CRU** (Schirmer et al., 2022) the hyperparameter space is as follows:

- The latent state dimension, with possible values [10, 20, 30].
- The number of basis functions, with possible values [10, 20].
- The bandwidth with possible values [3, 10].

## G    REGULARLY SAMPLED TIME SERIES DATA EXPERIMENTS

In this chapter, we show important findings related to generating **regularly sampled multivariate time series data** with `Physiome-ODE`. We discuss the generation process of 3 datasets that were chosen based on 2 important criterions. First is the previously mentioned JGD which was shown to be important indicator for the complexity of the dataset. Second criterion is for these datasets to include a high number of channels as we show through the following small experiment, how channel dependency matters more and have more effect when tested on our ODE-datasets compared to current benchmark datasets.

The datasets used here were DOK01, INA01, and HYN01 which all have a number of channels greater than 15 with 18, 29, and 22 channels respectively. Furthermore, they are ranked highly with respect to the JGD metric having values of 2,277, 2,218, and 1,548 respectively which makes them the most complex high dimensional (more than 15 channels) datasets in our benchmark. Datasets were generated based on the configuration at Appendix D, generating 100 samples for each dataset based on 100 different initial conditions that are drawn uniformly at random from a distribution based on the configuration given to the model. Each time series sample spans 300 timesteps. The datasets were split into a train/val/test split of 70/20/10 which is similar to the split done normally on the TSF literature for the weather, electricity and traffic datasets (Chen et al., 2023; Zeng et al., 2023; Nie et al., 2023).

We conduct experiments on state-of-the-art Time Series Forecasting models PatchTST (Nie et al., 2023), TSMixer (Chen et al., 2023), and DLinear (Zeng et al., 2023). For DLinear, there is only a univariate model (doesn't capture channel dependency), but for the other models, they can be implemented as univariate or multivariate (capture channel dependency) variants. So overall, we label the univariate models as **CI** (**C**hannel **I**ndependent), and multivariate models as **CD** (**C**hannel **D**ependent). These models are tested with the 3 ODE datasets mentioned earlier tuning the hyperparameters for the 3 models through random search of 20 trials for a predefined range of parameters.

For PatchTST, we set the following hyperparameter range:

- learning rate (range of floats) $\rightarrow [10^{-7}, 10^{-2}]$.
- number of encode layers (range of integers) $\rightarrow [1, 6]$.
- Dimensionality of fully connected layer (categorical) $\rightarrow [256, 512, 1024]$.
- Embedding dimension of the attention layer (categorical) $\rightarrow [128, 256, 512, 1024]$.
- dropout (range of floats) $\rightarrow [0, 0.9]$.
- fully connected layer dropout (range of floats) $\rightarrow [0, 0.9]$.
- patch size (categorical) $\rightarrow [4, 8]$.
- stride (categorical) $\rightarrow [2, 4]$.

For TSMixer, we set the following hyperparameter range:

- learning rate (range of floats) $\rightarrow [10^{-7}, 10^{-2}]$.
- number of blocks (range of integers) $\rightarrow [1, 5]$.
- hidden size of MLP layers (categorical) $\rightarrow [32, 64, 256, 1024]$.
- dropout (range of floats) $\rightarrow [0, 0.9]$.
- activation (categorical) $\rightarrow [ReLU, GeLU]$.

For DLinear, learning rate was tuned on the same range of values mentioned above. All models were tuned on the validation set. The results for the best performing model on validation set (lowest MSE error) were reported on the test set. We used a lookback window of 24 and forecasting horizon of 12 for all the conducted experiments on the ODE datasets. On the other hand, we report the results of the mentioned models on well-established evaluation datasets such as ETTh1, ETTh2, ETTm1, ETTm2, Weather, Electricity, and Traffic on a forecasting horizon of 96. The reported results from Table 6 give us interesting insights about our benchmark on the regularly sampled time series data.

Table 6: MSE Test loss reported on 3 ODE datasets as well as on 7 of the most popular regular TSF datasets. We highlight the best performing version (CI/CD) of each model in **bold**. The results were reported on 5 different baselines showing a clear trend of CD models being better on our generated ODE datasets. On the other hand, for benchmark datasets, this trend is not there as CI models are performing quite well.

| Dataset | PatchTST(CI) | PatchTST(CD) | TSMixer(CI) | TSMixer(CD) | DLinear |
|---|---|---|---|---|---|
| INA01 | 0.099 | **0.052** | 0.168 | **0.062** | 0.191 |
| DOK01 | 0.087 | **0.022** | 0.155 | **0.039** | 0178 |
| HYN01 | **0.001** | **0.001** | 0.019 | **0.002** | 0.046 |
| ETTh1 | **0.375*** | 0.416* | 0.361* | **0.359*** | 0.375* |
| ETTh2 | **0.274*** | 0.334* | **0.274*** | 0.275* | 0.289* |
| ETTm1 | **0.290*** | 0.326* | 0.285* | **0.284*** | 0.299* |
| ETTm2 | **0.165*** | 0.195* | 0.163* | **0.162*** | 0.167* |
| Weather | **0.152*** | 0.168* | **0.145*** | **0.145*** | 0.176* |
| Electricity | **0.130*** | 0.196* | **0.131*** | 0.132* | 0.140* |
| Traffic | **0.367*** | 0.595* | 0.376* | **0.370*** | 0.410* |

[*] Results are taken directly for PatchTST, TSMixer, and DLinear from the respective papers (Nie et al., 2023), (Chen et al., 2023), and (Zeng et al., 2023).

We notice from the results how unpredictable the results can be for the benchmark datasets, with **CI** models being better on some datasets and **CD** models being better on the rest. Also, one more interesting finding is how close the test loss is between a very simple CI model such as DLinear and the state-of-the-art models which can be concerning regarding how much channel dependency there is on the available benchmark datasets. On the other hand, for our ODE-generated datasets, the results are more consistent with clear better performance for **CD** models over their **CI** counterparts. Also, the best model overall was noticed to be always a **CD** model which shows the importance of capturing channel dependency on the ODE generated datasets. One more interesting finding is how worse is a DLinear model on our benchmark datasets where its results are at least 5 times worse than the best performing model. That shows how complex these datasets are, as well as how multivariate models can clearly benefit from intrinsic relation between the channels on such datasets.

# H LORENZ

We conducted a forecasting experiment on the chaotic Lorenz Attractor:

$$\frac{dx}{dt} = \sigma(y - x) \tag{40}$$

$$\frac{dy}{dt} = x(\rho - z) - y \tag{41}$$

$$\frac{dz}{dt} = xy - \beta z \tag{42}$$

Instead of varying the constants as we do in `Physiome-ODE`, we set the parameters to

- $\rho = 28$
- $\sigma = 10$
- $\beta = \frac{8}{3}$

We vary the sample the initial states from $x \sim [1, 3], y \sim [0, 2] z \sim [0, 2]$. Following Gilpin (2021), the task is to forecast the final $\frac{1}{6}$ of the IMTS based on the initial $\frac{5}{6}$. Since we always use the exact same constants, we create 200 instead of 2000 time series instances. Outside the mentioned changes, the experimental protocol is the same as the one used in our experiments with the ODEs from `Physiome-ODE`.

As shown in Table 7, there is no model which significantly outperforms the constant baseline GraFITi-C.

Table 7: Test MSE IMTS created on the chaotic Lorenz-Attractor.

| Dataset | GRU-ODE | LinODEnet | CRU | Neural Flow | GraFITi | GraFITi-C |
|---------|---------|-----------|-----|-------------|---------|-----------|
| Lorenz | 1.334±0.097 | 1.313±0.123 | 1.292±0.087 | 1.345±0.105 | 1.294±0.090 | 1.303±0.079 |

