# OpenReview forum: "Physiome-ODE: A Benchmark for Irregularly Sampled Multivariate Time-Series Forecasting Based on Biological ODEs"
_ICLR.cc/2025/Conference — ICLR 2025 Poster_

### Official Review · Reviewer_Sy9P · 2024-10-28

**Soundness:** 2
**Presentation:** 3
**Contribution:** 2
**Rating:** 3
**Confidence:** 4

**Summary:**

The paper presents a set of ODE generated benchmark dataset for Irregularly Sampled Multivariate Time-Series Forecasting task. Furthermore, it proposes a complexity metric (JGD) to evaluate the difficulty of different datasets. Finally evaluated common forecasting methods and a baseline predicting constant time series.

**Strengths:**

The paper presents a new set of baselines for Irregularly Sampled Multivariate Time-Series Forecasting, As there is a need for standard baseline in this area therefore the contribution is significant.
It also provides code to test existing methods on the benchmark, and introduces a new metric to assess the complexity of a given dataset. This metric correlates with the lowest prediction error achieved by the tested models.

**Weaknesses:**

The new benchmark is a standardized way of generating data from already public ODE models. This would still be valuable, but some of the choices are quite ad-hoc including the parameter noise and the initial condition selection.
These systems have significantly different sensitivity for different parameters and can change between operation regimes by tiny change of some parameters.  Currently the authors use a common $\sigma_{\text{const}}$ and hope for meaningful results. If this not happen (ODE explodes for example) the given time series is simply dropped. This may invalidate the benefit, that there is meaning behind these curated ODEs, by setting physically (or biologically in this case) nonsensical parameters or initial conditions. Expert knowledge should be used to decide what parameters should be fixed what can be modified, and sensitivity analysis should be carried out to create meaningful time series.

The mean gradient deviation (MGD) as metric of complexity is ad-hoc and questionable. As it is mentioned in the paper it automatically assumes faster oscillatory processes are less predictable. How about a huge (length of the rods is large, so in average slowly moving) double pendulum in its chaotic regime? Is it less complex than a very fast sine wave? Also systems like the Lorenz system changes between regimes (two “wings” of the “butterfly” ) suddenly but quite smoothly evolving in between, it seems this is not captured by the metric.
Correlation is found between the error and the composite metric (JGD) but it can be spurious, see bellow.

It is not clear from the paper how the training and the evaluation of the methods are carried out.  In the Experimental protocol (Line 351-352) It stands  “In our experiments models have to predict the last 50% of the time series after observing the first 50%.” How the ODE models were trained? the loss in train time was computed on similarly long sequence and backpropagated or it was trained on a single point future prediction task.
If the later that the methods had no fair chance to learn to, for example, fall back to the constant model, as they clearly trained optimizing short term error, and go out of phase. Due to the comparatively excellent result of the constant model, it needs to be cleared that this is not the case. In this regime fast oscillating systems are less predictable as well.

**Questions:**

1) Did you checked that parameters you modify by resampling from a distribution still meaningful? eg.: parameters have to be positive are positive, parameters have to have a given order (e.g. a > b always hold by biology) are such, logarithmic scale parameters are sampled in logarithmic scale etc.  Same for initial condition.
2) How the suggested complexity measure assesses Lorenz system for example?
3) You mention the necessity of normalizing the system values. How about time? What happens if a system timescale changed from measured in sec to measured in hours?
4) Describe the training protocol what is the loss computed on, how much ahead prediction is evaluated. Best if you create a figure, If it is changing for some methods, describe it specifically.

Minor:
I would consider calling the "ode constants", “ode parameters” as this is more frequently used, or mentioning at least both name, as it can be confusing.

---

> ### Author Response · Authors · 2024-11-21
>
> We want to thank the reviewer for the feedback and questions.
>
> To Q1:
> We do not guarantee that our parameter modifications are within natural bounds. However, we do not think that is extremely important for Machine Learning experiments. Ensuring that everything is within natural bounds/scales would necessitate enormous expert domain knowledge and manual effort. Consequently Physiome-ODE will contain some "unrealistic" samples.
>
> However, we do not think that this harms Physiome-ODE's ability to evaluate IMTS forecasting models.
>
> To Q2:
> The JGD for the Lorenz system is at 0.848, which is lower than the JGD of some of the ODE-systems contained, while the models show a higher MSE on this dataset. This finding is not really surprising as the Lorenz ODE is challenging not due to high frequency but due to unpredictable chaotic trajectories.
>
> To Q3:
> That should not change anything as we normalize the time to be in the range [0,1].
>
> To Q4:
> To clarify the training procedure we want to reemphasize that we create 2000 time series for each ODE system, which are generated using different ODE-constants and initial states (l.323). In each fold of the 5-fold crossvalidation we split these 2000 time series into train(1400) validation (400) and test (200). Finally we split each time series into observation range and forecasting range at 50% of the total time horizon. Each model is trained to predict the targets (in the forecasting range) based on the observations (in the observation range). The loss is computed based on the prediction of the forecasting targets.  This procedure is following the existing IMTS forecasting literature [1,2,3]
>
> References
> [1] De Brouwer, Edward, et al. "GRU-ODE-Bayes: Continuous modeling of sporadically-observed time series." _Advances in neural information processing systems_ 32 (2019).
>
> [2] Biloš, Marin, et al. "Neural flows: Efficient alternative to neural ODEs." _Advances in neural information processing systems_ 34 (2021): 21325-21337.
>
> [3] Yalavarthi, Vijaya Krishna, et al. "GraFITi: Graphs for Forecasting Irregularly Sampled Time Series." _Proceedings of the AAAI Conference on Artificial Intelligence_. Vol. 38. No. 15. 2024.

---

> > ### Comment · Reviewer_Sy9P · 2024-11-25
> > **Reply to Author comments**
> >
> > Thank you very much for your reply.
> >
> > The authors clarified some questions about the training procedure, but reinforced me in my view that the JGD metric is actually inappropriate for measuring hardness of a task.
> >
> > About Q1: I have to partly disagree, it is clearly part of the paper's claim that the models are meaningful. Even the name of the dataset focusing on that. This benefit is reduced by using nonsensical parameters. I agree that curating everything is a large effort, and I also agree that this does not make the dataset useless for its purpose, but it clearly reduces its value.

---

### Official Review · Reviewer_nAcA · 2024-11-02

**Soundness:** 2
**Presentation:** 2
**Contribution:** 1
**Rating:** 3
**Confidence:** 4

**Summary:**

This paper introduces a new benchmark for multi-variable time series (IMTS) prediction called Physiome-ODE, which is generated based on biological dynamical equations (ODEs). The current evaluation methods for irregularly sampled and missing-value time series prediction mainly rely on limited datasets, which may not accurately assess the performance of models due to their small size and diversity. The authors develop a new methodology to generate and filter challenging IMTS datasets from ODEs, successfully creating a significantly larger and more diverse benchmark than existing evaluation settings. Physiome-ODE consists of 50 independent datasets derived from ODE models used in biology research over decades. By comparing the performance of existing IMTS prediction models on this new benchmark, the authors reveal different strengths among the models and indicate that some current prediction models can demonstrate stronger abilities on Physiome-ODE compared to traditional evaluation datasets. Additionally, the paper proposes a new metric, Joint Gradient Deviation (JGD), to measure the difficulty of datasets, demonstrating that the benchmark can effectively distinguish between datasets and models of different complexities. The introduction of Physiome-ODE not only provides a more comprehensive and realistic assessment platform for IMTS prediction but also promotes future research in this field.

**Strengths:**

- Physiome-ODE provides a larger and more diverse benchmark, allowing for a more comprehensive assessment of IMTS prediction models.

- By comparing existing models' performance on Physiome-ODE, it is possible to discover some models with stronger abilities in handling irregular sampling and missing values.

- A new metric called Joint Gradient Deviation (JGD) has been proposed, which measures the difficulty of the dataset and helps distinguish between different complexity levels of datasets and models.

**Weaknesses:**

- The contribution of the proposed Physiome-ODE dataset is not clearly articulated in the manuscript, making it difficult to understand its unique advantages compared to existing time series forecasting benchmarks.

- The study only considers ODE-based predictive models and overlooks more recent models, such as TimeMixer and TimesNet. This limited model selection restricts the breadth of the comparison, potentially missing insights that could be gained from newer approaches.

- Although the study highlights that datasets created with Physiome-ODE encourage models to learn channel dependencies, it does not explain why channel-independent models like PatchTST, DLinear, PDF, and SparseTSF are observed to perform better on traditional datasets. This lack of explanation, coupled with the absence of supporting experiments, leaves the findings incomplete and reduces clarity on model performance differences.

**Questions:**

- Why is Joint Gradient Deviation (JGD) introduced, and what specific advantages does it offer in creating benchmarks?

- The authors do not adequately address how the Physiome-ODE dataset ensures representativeness and reliability. Without a clear explanation of its distinctive features and validation processes, the dataset's credibility as a benchmark for time series prediction remains uncertain.

- 5-fold cross-validation is used for model evaluation. Is this partitioning sufficiently representative of the dataset's diversity? Could the way the data is split introduce any biases in the results?

- By focusing solely on ODE-based models, the authors fail to incorporate recent advancements like TimeMixer and TimesNet, which may offer alternative or improved performance. This omission results in an incomplete evaluation, as it leaves out potentially competitive models that could impact the study's findings.

- The authors do not clarify why channel-independent models (e.g., PatchTST, DLinear, PDF, SparseTSF) are reportedly more effective on standard datasets, nor do they provide experimental evidence to support this observation. Without a clear rationale or relevant experiments, the paper’s insights into channel dependencies remain unsubstantiated, limiting the strength of its conclusions. Furthermore, this work does not consider the impact of data stationarity on the results.

---

> ### Author Response · Authors · 2024-11-21
>
> We thank the reviewer for valuable suggestions and constructive feedback.
> However we want to clarify a few things:
>
> To W2 and Q4:
> Our study focuses on Irregularly sampled multivariate time series (IMTS) with missing values. TimeMixer and TimesNet are both models designed for regularly sampled time series, which is why we did not included these models in our main experiment.
>
> To W3 and Q5:
> The fact, that the channel-independent models are more effective on standard datasets, is clearly indicating that channels contained in these datasets are rather independent.  For example, PatchTST gains no advantage in modeling channel dependencies as these seem to not carry any useful information and the additional model-complexity leads to overfitting.
> On INA01 and DPL01 however, we can observe the opposite. Here, PatchTST actually benefits from modeling channel depencies.
>
> To Q1:
> JGD was introduced so we could filter and configure the ODE systems from Physiome in a systematic manner. We wanted to leverage all the ODEs published on this website to automatically create datasets, which can be used for IMTS forecasting experiments. Therefore, we needed a metric to find how well-suited an ODE system will be for our benchmark and we came up the JGD, to discriminate datasets which are too simple to forecast. We could show in our experiments that the JGD-metric is fulfilling its purpose adequately.
>
> To Q2:
> We want to refer to our answer to Question 1 of reviewer Sy9P.
>
> To Q3:
> We opted for 5-fold cross validation following the IMTS forecasting literature [1,2,3].
> Different validation protocols could be valid for Physiome-ODE. For example, one could have completely different generated time series in every fold. Nevertheless, there is no reason for 5-fold-cross validation being insufficient.
>
>
> References
> [1] De Brouwer, Edward, et al. "GRU-ODE-Bayes: Continuous modeling of sporadically-observed time series." _Advances in neural information processing systems_ 32 (2019).
>
> [2] Biloš, Marin, et al. "Neural flows: Efficient alternative to neural ODEs." _Advances in neural information processing systems_ 34 (2021): 21325-21337.
>
> [3] Yalavarthi, Vijaya Krishna, et al. "GraFITi: Graphs for Forecasting Irregularly Sampled Time Series." _Proceedings of the AAAI Conference on Artificial Intelligence_. Vol. 38. No. 15. 2024.

---

> > ### Comment · Reviewer_nAcA · 2024-11-22
> >
> > Thanks for your kind response. However, the replies do not sufficiently address the concerns raised, particularly in providing detailed mechanisms and supporting analyses.
> >
> > - More detailed analysis and experimental evidence are needed to substantiate the utility of JGD.
> >
> > - Further details on dataset construction, model diversity, and coverage of prediction challenges are essential to establish the credibility.
> >
> > - Including more recent models could provide a more comprehensive evaluation and uncover additional insights.
> >
> > - A deeper investigation into the performance of channel-dependent and channel-independent models, as well as the influence of data non-stationarity, is necessary.
> >
> > These responses fail to resolve the concerns. Therefore, I will maintain my original rating.

---

### Official Review · Reviewer_ouEs · 2024-11-03

**Soundness:** 2
**Presentation:** 2
**Contribution:** 2
**Rating:** 3
**Confidence:** 3

**Summary:**

An irregularly sampled multivariate time series (IMTS) forecasting benchmark called "Physiome-ODE," which is derived from biological ordinary differential equations (ODEs), is proposed in this paper. Through the provision of a more extensive and varied collection of datasets, it seeks to overcome the shortcomings of the existing IMTS benchmarks, which are small and unvarying. The authors present Joint Gradient Deviation (JGD) as a metric to evaluate the complexity of datasets, asserting that it gives the benchmark a significant degree of rigor.

**Strengths:**

- Physiome-ODE offers a novel approach to IMTS benchmarks by generating datasets using ODEs, which is an advancement over the few IMTS datasets currently available. The idea of creating datasets using biological ODEs may benefit the scientific community.

- The paper offers a rigorous mathematical setup, especially when defining the JGD metric and the IMTS problem.

- This paper has a wider empirical foundation because it includes experiments on 50 datasets. This could be a benefit when evaluating model performance variability.

**Weaknesses:**

- The theoretical explanation of JGD and how it relates to dataset complexity is unclear and excessively detailed. It is difficult to assess the reliability of the claims due to the heavy reliance on mathematical notation without adequate intuitive explanation.

- Although the authors assert that JGD scales super-exponentially with the Lipschitz constant, they offer no supporting data or examples. Furthermore, without a comparison to other well-known metrics like variance or entropy, it is difficult to determine how useful JGD is as a complexity metric.

- The robustness of the results under different experimental conditions is called into question because there is no sensitivity analysis on the numerical solver or noise parameterization. Additionally, there is limited discussion on the generalizability of Physiome-ODE beyond biological applications.

**Questions:**

- The integration of observation noise into this configuration is not adequately explained by Equation (2). While Equation (3) discusses the addition of noise to the generated IMTS data later on, but there is no obvious connection to Equation (2), so it is unclear how the noise is actually added in practice. The data generation process might be more rigorous if the differential equation were explicitly formulated to account for noise, as would be the case with a stochastic differential equation (SDE) framework.

- Simple statistical models (like linear regression and ARIMA) and more complex models (like neural SDEs) are conspicuously absent from the selection of baseline models, which lacks rigor.

- The existence and uniqueness of the solution are not discussed, particularly in Equation (12). The robustness of this optimization is doubtful in the absence of conditions guaranteeing that a unique maximum exists for JGD over these parameters. For example, what are Equation (12)'s spread parameter bounds?

- Can the authors substantiate their assertion that JGD scales super-exponentially with the Lipschitz constant with empirical data or an example?

- Lemmas 1 and 2 (Appendix B) are used to approximate MGD and MPGD with finite samples, but the assumptions on function continuity and differentiability are not stated clearly.

---

> ### Author Response · Authors · 2024-11-21
>
> We thank the reviewer for the constructive review and questions.
>
> To W1:
> We designed the JGD to be a simple metric that helps us to automatically select good continuous ODE systems based on numerical samples. As we describe in Section 4 MGD is deviation of Gradients within one channel and the MPGD is the devation of gradients in between channels, while the JGD is simply the product of MGD and MPGD. We do not think that our definition is excessive as we devoted only half a page for the description of JGD and half a page to its computation based on discrete samples.
>
> To W2 and Q4:
> The fact that the MGD grows super-exponentially with the Lipschitz constant is just a minor and theoritical finding which is why it is described in the appendix. It is not clear how this theoretical finding could be supported with data.
>
> To W3:
> We agree that a sensitivity analysis on the numerical solver would be useful and leave that for future work. We do not see any reason why our datasets would be less generalizable than any other dataset. However it is not clear how would evaluate generalizability of a dataset/benchmark, which is why we are unsure on which points one could base such a discussion.
>
> To Q1:
>
> As stated in L.328 we add Gaussian noise with a variance of 0.05.
>
> To Q2:
>
> ARIMA cannot be applied to IMTS data. CRU are an advanced version of SDE.
> The models we selected for this work serve the purpose, that they show that Physiome-ODE solve the problem that we outlined in section 3
>
> To Q3:
>
> For our work the existence and uniqueness of optimal parameters described in eq. 12 are actually irrelevant. As ODE-models with extremly high JGD's are actually not well-suited to bench machine learning models. E.g. ODE-constants that lead to "exploding ODEs" would have insanely high JGD values, as described in l. 331f. Instead we optimize the JGD by varying the constants in a very limited space as described in 315f. and exclude any configurations that lead to exploding ODEs.
>
> To Q5:
>
> We do not find any unclear statements in our proof. Could you point out the assumptions you are referring to ?

---

> ### Comment · Reviewer_ouEs · 2024-11-24
>
> Thank you for the detailed responses to my questions.
>
> - The connection between JGD and dataset complexity remains insufficiently intuitive for a broader audience. While you emphasize that JGD is designed to be a simple and practical metric, the explanation provided in the paper still feels unclear.
>
> - While I understand `the sensitivity analysis on the numerical solvers` was not a focus of your work, the lack of this analysis leaves open questions about the robustness of the results. For a benchmark to be widely adopted, robustness to implementation details such as solvers could be a critical factor.
>
> - Conventional method (such as ARIMA, GRU) can be applied by simple imputation (such as mean-imputation, see [1]). It may shows poor performance, but can emphasize the difficulty of the problems. Also, Neural SDE based methods are suggested to handle forecasting and further tasks in IMTS data [2]. I recommend you to check further related studies considering real-world dataset for the IMTS forecasting task.
>
> > [1] Che, Z., Purushotham, S., Cho, K., Sontag, D., & Liu, Y. (2018). Recurrent neural networks for multivariate time series with missing values. Scientific reports, 8(1), 6085.
>
> > [2] Oh, Y., Lim, D., & Kim, S. (2024), Stable Neural Stochastic Differential Equations in Analyzing Irregular Time Series Data, The Twelfth International Conference on Learning Representations (ICLR) 2024.
>
> - My concern lies in the implicit assumptions about continuity and differentiability of the functions involved in approximating MGD and MPGD. Explicitly stating these assumptions in the paper would strengthen the theoretical rigor.

---

### Official Review · Reviewer_q9fn · 2024-11-03

**Soundness:** 3
**Presentation:** 2
**Contribution:** 3
**Rating:** 6
**Confidence:** 3

**Summary:**

The paper introduces Physiome-ODE, a novel benchmark for irregularly sampled multivariate time series (IMTS) forecasting derived from ordinary differential equations (ODEs). Physiome-ODE consists of 50 individual datasets created using biological ODE models. The authors highlight the need for a more challenging and biologically relevant benchmark for IMTS, as existing benchmarks primarily rely on only a few datasets and even simple constant-value baselines outperform complex ODE-based methods. Using Joint Gradient Deviation (JGD) as a metric, they select challenging ODE instances from the Physiome Model Repository, ensuring that the benchmark captures diverse levels of complexity. The paper also provides a comprehensive evaluation of state-of-the-art forecasting models, comparing methods based on neural ODEs with simpler, non-ODE methods.

**Strengths:**

1. Significant Contribution to IMTS Benchmarking: The introduction of Physiome-ODE represents an important step forward in providing a robust and biologically relevant benchmark for irregular time series forecasting, filling a notable gap in the current research landscape.

2. Use of JGD for Dataset Complexity: The introduction of Joint Gradient Deviation (JGD) to measure the gradient variance and dataset complexity is a well-justified and creative way to ensure that the generated datasets vary in difficulty, addressing the shortcomings of existing IMTS datasets.

3. Broad and Diverse Dataset Generation: The benchmark is derived from biological ODEs, which are inherently multivariate and often irregularly measured. This connection to real biological processes makes Physiome-ODE highly relevant for practical forecasting applications, especially in healthcare and biology.

4. Detailed Evaluation of State-of-the-Art Methods: The evaluation results indicate the diversity of the Physiome-ODE benchmark, where different models excel in different scenarios, highlighting no single model as the best for all datasets. This realistic scenario is useful for researchers to understand the strengths and weaknesses of existing methods.

**Weaknesses:**

1. The paper could benefit from a more detailed comparison against existing benchmarks for IMTS forecasting. While the authors do compare some models to existing datasets (such as MIMIC-IV and PhysioNet), a direct comparison of Physiome-ODE’s added value over these datasets using a common evaluation metric would be more convincing.

2. The majority strategy for selecting challenging ODE instances, although effective in finding complex trajectories, might overlook personalized or localized causal differences that are crucial for domains such as personalized medicine. This lack of granularity could limit the applicability of Physiome-ODE to more individualized forecasting tasks.

3. Creating and using Physiome-ODE is computationally intensive, especially with diffusion-based data generation and the JGD optimization steps. The paper lacks an analysis of how the dataset's computational demands impact its usability, particularly for researchers with limited access to high-performance computing resources.

4. Physiome-ODE is a semi-synthetic benchmark, as the original biological datasets are often not publicly available. This limits the interpretability and direct clinical relevance of the benchmark since it relies on models rather than real patient data. A more comprehensive discussion on the implications of using purely ODE-generated data, including potential biases, would be beneficial.

**Questions:**

1. How feasible is it for other researchers to replicate Physiome-ODE in environments with limited computational resources?
2. Could the proposed dataset generation method be adapted to capture personalized features in time series, such as patient-specific characteristics?
3. How does Physiome-ODE perform against benchmarks like Monash Time Series Archive or PDEBench in terms of practical outcomes for IMTS forecasting?
4. Given that the generated datasets are semi-synthetic, how closely do the generated ODE solutions resemble actual biological processes observed in real data? Would you suggest any metrics to measure the closeness?

---

> ### Author Response · Authors · 2024-11-21
>
> We want to thank the reviewer for the detailed feedback.
>
> To W1:
> We agree that such a metric would give additional evidence that Physiome-ODE is superior to existing evaluation datasets. Is there any specific metric that you have in mind ? Currently, we support our claims with the relative success of our constant baseline (GraFITi-C), the number of included datasets and the fact that the relative performance of models changes over datasets.
>
> To W2:
> Our approach was to create a large benchmark for IMTS forecasting in a systematic and automated manner. This inherently will cause a certain lack of granularity. To find the model for an individual forecasting application, we recommend to use data from the respective domain. Physiome-ODE is designed to support IMTS modeling research in general and covers a broad range of dynamics and patterns.
>
> To W3 and Q1:
> Actually, the creation of Physiome-ODE is computationally cheap. The ODE-systems we use for the creation varies in complexity and therefore each dataset needs different amount of time to be created. The creation of each dataset finished in 30mins- 12h and we used the CPU's from our computing cluster, where the strongest CPU is the AMD EPYC 7713P and the weakest one was the Intel E5-1620v4.
>
> To Q2:
> Yes that should be really feasible with the code provided from us, as patient-specific characteristic will somehow result in certain ODE parameters and initial states.
>
> To Q3:
> We have not made such an experiment. One could use the Monash datasets and create IMTS from them by sampling in a similar as it was previously done with USHCN. However this is not promising as the vast majority of Monash datasets are univariate or have multiple independent channels and therefore not interesting for IMTS research.
>
> To Q4:
> One could easily compute an MSE of the generated ODE solutions and the actual measurements. However, we do not have access to these. We assume that the ODE models created from the biological researchers are highly accurate and close to measurements.

---

> > ### Comment · Reviewer_q9fn · 2024-11-25
> >
> > Thank you for the responses and they clarified most of my concerns.

---

### Meta-Review · Area_Chair_KZjZ · 2024-12-29

**Metareview:**

This paper introduces a large benchmark comprised of 50 datasets for irregularly sampled time series. The paper provides a metric for assessing dataset complexity that some reviewers have judged not fully justified. Although most of the reviewers have recommended rejection, I recommend this paper to be accepted to the conference. The field of time series modeling needs benchmarks and this paper contributes differently than Monash, PDEBench, and other existing time series benchmarks used in the literature. I recommend the authors to add a more detailed justification of the metric for assessing hardness.

**Additional Comments On Reviewer Discussion:**

The authors provided a rebuttal, answering the questions of the reviewers. I believe the rebuttal addresses most of the reviewers' concerns.

---

### Decision · Program_Chairs · 2025-01-22

Accept (Poster)